# Nanostructures as indicator for deformation dynamics

Sarah Incel[1] ✉, Markus Ohl[2,5], Frans Aben[3], Oliver Plümper ![ORCID][2,6,7] &
Nicolas Brantut[1,4]

We determine the feedback between fault dynamics and fault gouge structures by examining gouge structures that form during rupture and slip of initially intact granite under upper crustal conditions. Experiments were conducted under quasi-static ($3 \times 10^{-5}$ mm/s), weakly dynamic (0.27 mm/s) and fully dynamic (≫1.5 mm/s) slip conditions, with or without fluids, and limited slip displacement (max. 4 mm). The extent in gouge amorphization positively correlates with deformation rate, and we detect evidence of melting, e.g., magnetite nanograins, associated with the highest deformation rates. Gouge nanostructure is directly correlated to power dissipation rather than total energy input. The presence of amorphous material has no detectable impact on the strength evolution during rupture. We highlight that gouge textures, generally associated with large displacements and/or elevated pressure and temperature conditions, can form during small slip events (Mw < 2) in the upper crust from initially intact materials.

The formation of tectonic faults and the accumulation of shear displacement across them is associated with profound microstructural transformations of the rocks forming their core, potentially accommodating very large strains. Under upper crustal conditions, fault formation and slip typically produce a gouge (e.g. refs. 1–4 and references therein). Depending on fault dynamics (quasi-static or rapid earthquake slip), specific localised transformations may occur: extreme grain size reduction, amorphization, dehydroxylation of hydrous phases, and frictional melting. Only a small subset of those transformations is conserved in the geological record and is an unambiguous marker of seismic slip in natural microstructures[5,6].

Furthermore, the localised transformations of fault zone materials might be linked to changes in fault strength in the short- and long-term. For instance, theoretical studies have shown that the degree of strain localisation within a fault gouge has a first order impact on the weakening behaviour at seismic slip rates (e.g. refs. 7–10). The formation of amorphous silica has been shown experimentally to produce dynamic weakening (e.g. refs. 11–13). Similarly, extreme grain size reduction has been linked to fault weakening (e.g. refs. 14–18). The most extreme, and perhaps the most recognisable fault structure developed at high slip rate is the formation of frictional melt, called a pseudotachylite once quenched, which causes dramatic weakening as soon as melt patches form a continuous layer across the fault (e.g. refs. 19–22). In nature, grain size reduction, amorphization and melting generate highly reactive materials, prone to recrystallisation and alteration, leading to the overall maturation of fault zones[23].

One key aspect of fault evolution during slip is the transition from microfracturing and grain size reduction, i.e. gouge formation (e.g. refs. 24–26), to either mechanical amorphization[27] and/or frictional melting. At high slip rates in granitic gouges, these processes appear sequentially with increasing slip (e.g. ref. 28), and the onset of melting is marked by an initial strengthening, i.e. a "viscous break"(e.g. refs. 29,30), and subsequent weakening. At low pressure (up to 10 to 20 MPa), this entire sequence requires around 1 m of total slip, and the weakening is correlated to power dissipation[31,32]. However, the production of amorphous materials with microstructures reminiscent of

[1]GFZ Helmholtz Centre for Geosciences, Potsdam, Germany. [2]Utrecht University, Utrecht, The Netherlands. [3]TNO, Utrecht, The Netherlands. [4]University College London, London, UK. [5]Present address: Oxford Instruments GmbH, Wiesbaden, Germany. [6]Present address: Faculty of Geosciences, University of Bremen, Bremen, Germany. [7]Present address: MARUM - Center for Marine Environmental Sciences, University of Bremen, Bremen, Germany. ✉e-mail: sarah.incel@gfz.de

pseudotachylites has also been documented at much smaller strains, slower deformation rates ($<10^{-5}$ s$^{-1}$) and higher pressures (1 GPa; Pec et al.[33,34], which is apparently controlled by total energy dissipation[35]. Amorphization is also produced by grinding at low stress and large strain[27]. Thus, in crustal rocks, there seems to be a wide range of conditions that can lead to amorphization and pseudotachylite-like structures, e.g. flow textures, which has important consequences for the detection of geological markers to estimate paleo fault dynamics. Furthermore, the presence of amorphous material and frictional melt changes the long-term strength of faults[36–38], and it is thus crucial to determine how those materials form.

Most of the aforementioned structures and their impact on fault strength have been well documented in laboratory experiments conducted in rock samples with pre-existing faults, either along engineered (sawcut) surfaces or in artificial fault gouges, undergoing typically large slip displacements (from several mm up to several tens of m). However, the state of natural faults prior to fault slip is likely more complex than represented in friction experiments: Faults are often cohesive due to healing and sealing processes through chemical reactions[39,40], and present roughness, which may impact the development of microstructure and the associated weakening processes. It has been shown experimentally that such processes can cause the full strength recovery of rock samples (e.g. ref. [15]), leading to the conclusion that the nucleation of earthquakes as well as the associated unstable slip propagation may be better analysed in terms of fracture dynamics of intact rocks rather than by frictional properties of faults (e.g. refs. [41,42]).

Here, we investigate micro- and nanostructures, which form during faulting of initially intact granite, to determine their relationship to the dynamics of the faulting process. Using intact rock samples as starting material allows us to explore the role of the initial shear fault formation, spontaneous strain localisation and slip along naturally rough faults on microstructure development, which has not been investigated in any systematic way since the 1970's[43–45]. We detect clear evidence of amorphization in all experiments, and find unambiguous evidence of gouge melting in the dynamically deformed sample. Amorphization does not seem to be linked to any particular weakening behaviour. Our observations show that faulting in intact rock, e.g. along immature faults, with limited slip (equivalent to magnitude <2 earthquakes) at shallow depth (a few km) already produces evolved textures, some of which could remain as geological markers of earthquake slip.

## Results

### Mechanical data

To study the influence of rupture and slip rates on fault gouge structure in initially intact granite, it was crucial to control and vary rupture and slip rates between laboratory tests. The effective pressure, defined as $P_{eff} = P_c - P_f$ with $P_c$ being the confining pressure and $P_f$ denoting the pore fluid pressure, was kept constant at 40 MPa throughout the experimental set, yet $P_c$ and $P_f$ differed between tests (Fig. S1). Variations in slip rate were achieved by following three different experimental procedures. First, through cycles of loading and unloading guided by the detection of acoustic emission to obtain controlled and slow failure resulting in an average slip rate of around $3 \times 10^{-5}$ mm/s (sample WG06). Second, by deforming the sample under high pore fluid pressure conditions, i.e. 70 MPa, resulting in a feedback between slip-induced dilation, fluid vaporisation, and the stabilisation of rupture during failure leading to a slip rate of 0.27 mm/s (sample WG12; Fig. S1; Brantut[46], Aben and Brantut[47]. Third, through unsupervised rupture under either dry conditions or under low pore fluid pressure conditions, i.e. 20 MPa, which results in dynamic failure of the samples and slip rates $\gg$1.5 mm/s under either dry (WG N03) or wet (WG14) conditions (Fig. S1; Brantut[46], Aben and Brantut[47]. The estimated dynamic slip rates are lower bounds as failure occurred faster than the recoding rate of the stress measurement, indicated by the dashed lines in Figs. 1 and S1b. To facilitate the correlation between sample names and their respective failure durations or slip rates, we will refer to the sample WG06 that failed slowest as controlled sample. Sample WG12 will be called self-stabilised sample, and samples WG14 and WGN03, which failed dynamically in less than a second, will be denoted as wet dynamic and dry dynamic, respectively.

Prior to failure, all samples reached a differential stress of around 380 MPa, except for the sample deformed under dry conditions that failed rapidly in a fully dynamic manner (sample WG N03; Fig. 1a). The latter attained a higher peak stress of around 560 MPa, because the drill core was not initially notched (see Experimental methods). Total accumulated slip was estimated to range between 1.2 and 4.2 mm (Fig. 1b). All initially notched samples show a similar evolution of apparent friction or ratio between shear and normal stress until around 0.7 mm of equivalent slip, beyond which the wet dynamic sample failed dynamically (indicated by the dashed line in Fig. 1c). The controlled sample reached a residual shear strength of $\approx$80 MPa, equivalent to a friction coefficient of 0.89, and together with the self-stabilised sample, they reached a similar apparent friction of around 0.9 (Fig. 1b, c).

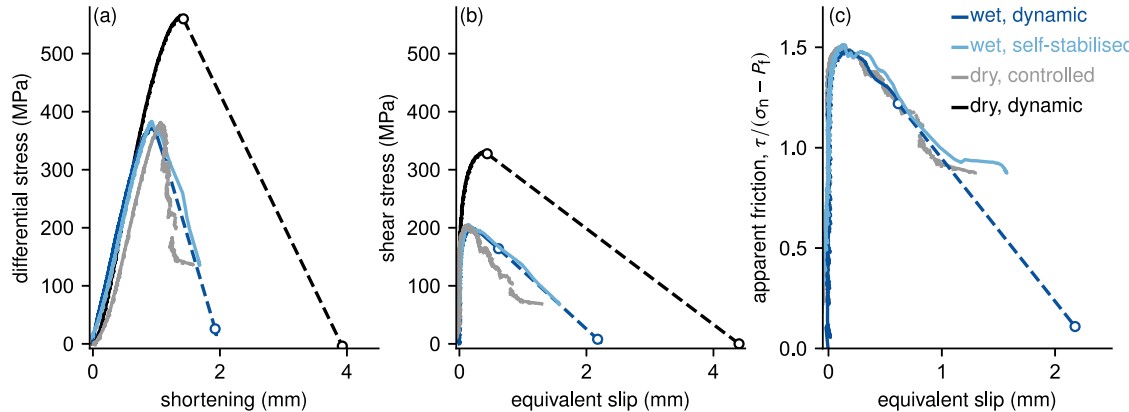

**Fig. 1 | Mechanical data from all deformation tests. a** Differential stress vs. shortening, **b** shear stress vs. equivalent slip, and **c** apparent friction vs. equivalent slip, for the four triaxial compression tests conducted on initially intact Westerly granite samples. All samples were deformed under the same effective pressure of 40 MPa. The wet dynamic sample was under a confining pressure of 60 MPa and a pore fluid pressure of 20 MPa. For the wet self-stabilised sample we went to a confining pressure of 110 MPa and a pore fluid pressure of 70 MPa. Please see the main text and the Methods section for further details on the mechanical data.

## Microstructural analyses

In order to investigate a potential influence of peak slip rate on fault gouge microstructures, we analysed each sample's fault gouge structure by using a scanning electron microscope (SEM) in backscattered-electron mode (BSE). At low magnification, the microstructures of the four different samples all look relatively similar, showing extensive grain size reduction in their respective fault zones (Fig. 2a, c, e, g). Increasing the magnification, we observe that all fault gouges contain grains ranging in size from sub-$\mu$m to a few tens of $\mu$m (Fig. 2b, d, f, h). In every fault gouge, we find zones in which we cannot distinguish between individual grains—even at the highest magnification at the SEM. We assume that these sites potentially contain amorphous material. These potentially amorphous zones appear more frequent in the samples that failed dynamically than in the other samples that failed slowly in either a self-stabilised or a controlled way (Figs. 2b, d, f, h and S2).

Flakes of submicron-sized mica produce textures that resemble flow textures in the controlled and the self-stabilised samples (WG06 and WG12), respectively (Figs. 2d and S2a, b). Much more striking flow textures can be found in fault gouges of the samples that failed rapidly in a dynamic manner (WG14 and WGN03; Figs. 2f, h and S2c, e). Furthermore, these zones are often associated with vesicles (Figs. 2f and S2h). Besides the apparent chemical heterogeneity enforcing the effect of a flow texture, we find several grains that appear to be embedded within these potentially amorphous zones. These grains, which range in size, are frequently associated with schlieren textures and exhibit strongly sutured grain boundaries (Figs. 2h and S2e). Another observation, restricted to the dynamically failed samples (WG14 and WGN03), is the presence of numerous grains of only a few 10s to 100s of nm in size—called nanograins in the following—that appear bright in BSE mode relative to the surrounding material (Fig. S2e, f). These nanograins are far smaller than the spot size used for energy-dispersive spectroscopy (EDS) at the SEM, i.e. around 1 $\mu$m, leading to mixed chemical analyses. Yet, it seems that they are rich in Fe, matching the observed brightness contrast between grains and the surrounding material. Regardless of the presence or absence of a pore fluid, we find flow textures, vesicles, and nanograins in both dynamically failed samples.

To investigate the nanostructures of the samples, focused-ion beam (FIB) sections were cut in selected areas (Figs. 2b, d, f, h and S2c, h). Since the FIB samples are small, and melting as well as grain comminution are material-specific processes, we decided to sample areas that are most representative for the entire fault gouge. Because felspar is the most abundant mineral in Westerly granite, we selected feldspar-rich areas to further investigate the samples' respective nanostructures.

## Nanostructural analyses

The nanostructure of the controlled sample (WG06) shows a broad range in grain size (Fig. 3a). Most larger grains, i.e. >200 nm, are cracked whereas smaller grains are not (Fig. 3a, b). The smallest individual grains we found are fragments of a few nm in size (Fig. 3b). The material surrounding these grains was identified as epoxy resin due to the absence of oxygen in element distribution maps, using EDS. The selected area electron diffraction (SAED) patterns I to III in Fig. 3 exhibit numerous clear diffraction spots, which are indicating the presence of crystalline material, as well as a faint halo, produced by diffraction of non-crystalline or amorphous material.

In the nanostructure of the self-stabilised sample (WG12), we observe layers caused by variations in brightness contrast in high-angle-annular dark-field (HAADF) mode, reflecting differences in chemical composition (Fig. 4a). Using EDS, these different layers were identified as a feldspar-rich layer, a quartz-rich layer, and a biotite-rich layer. The SAED pattern I, which we took within the feldspar-rich layer, shows numerous clear diffraction spots as well as a diffuse halo

(Fig. 4a). A bright field (BF) mode image exhibits several grains surrounded by a material that shows no sign of crystallinity (Fig. 4b). The adjacent quartz layer predominantly exhibits angular grains of a few 10s to 100s of nm in size (Fig. 4), and the SAED of this layer, SAED II, shows several clear diffraction spots as well as a faint diffuse halo. Numerous biotite flakes produce a diffraction pattern that resembles a powder diffraction pattern, in which small and randomly oriented grains form Debye rings (SAED III; Fig. 4).

The wet dynamic sample (WG14) exhibits fragments that float within a material exhibiting no indication of crystallinity (Figs. 5 and S2d; S3). Within this apparently amorphous material, we find smaller grains as well as vesicles that differ significantly in size and shape, i.e. from almost spherical or lens-shaped to more complex shapes (black arrows in Fig. 5). Diffraction pattern taken of different zones within the fault gouge, either reveal numerous clear diffraction spots and a diffuse halo (SAED I), almost exclusively a diffuse halo (SAED II and III), or a diffraction pattern revealing clear and strong diffraction spots arranged in a periodic manner (SAED IV; Fig. 5). Using the SAED pattern IV we were able to identify this grain as orthoclase, revealing lenticular shaped features (white arrow in Fig. 5). These features exhibit the same or a similar brightness contrast relative to the material surrounding this grain (white arrows in Fig. 6). Orthoclase grains often show sutured grain boundaries (white arrows in S2d; S3d).

The fault gouge of the dry dynamic sample (WGN03), is almost completely composed of material that does not show any sign of crystallinity, reflected by the exclusive appearance of diffuse halos in the SAED patterns I and II in Fig. 7a. However, we observe numerous bright grains (in HAADF mode) that exhibit a similar grain size of a few tens of nm. Combining the structural data from the SAED together with chemical information from the element distribution maps enabled us to identify this phase as magnetite (Figs. 7b, c and 8d–g). These magnetite grains appear euhedral and sometimes show concave crystal faces (Figs. 7b, c and 8c, d). Element distribution maps show a clear depletion in Fe, O, and Ti, and an enrichment in Si in the material immediately adjacent to the magnetite grains (Figs. 7c and 8e, f). The nanostructure of the fault gouge zone that cuts right next to a biotite grain exhibits large vesicles (Fig. S2g, h). Biotite fragments, present in the nanostructure, show numerous subparallel features that are all lens-shaped (red arrows in Fig. 8b).

## Discussion

### Influence of deformation dynamics on fault gouge structure

The SAED patterns together with BF images, exhibiting the presence of grains smaller than 10 nm (Figs. 3b and 4b), confirm the presence of amorphous material in every fault gouge (Figs. 3, 4, 5 and 7). By qualitatively assessing the amount of amorphous material from the intensity of the diffuse halos and the number of clear diffraction spots in the SAED patterns, we find a positive correlation between peak slip rate and the amount of amorphous material present in the sampled areas with an increase in amorphous material as slip rate increases (Fig. 9).

### Work vs. power

To assess whether variations in slip rate are the underlying cause for the observed variations in fault gouge structure, we calculated and compared the total energy dissipated, i.e. the work, during each test with the calculated power, i.e. the work rate, expended per unit area. To estimate how much energy was dissipated to obtain the different fault gouges, we used the relation

$$W_{\text{tot}} = \int_0^{\delta_{\text{final}}} \tau d\delta, \tag{1}$$

where the total energy density or total work ($W_{\text{tot}}$) can be expressed by integrating the measured evolution of shear stress ($\tau$) over the

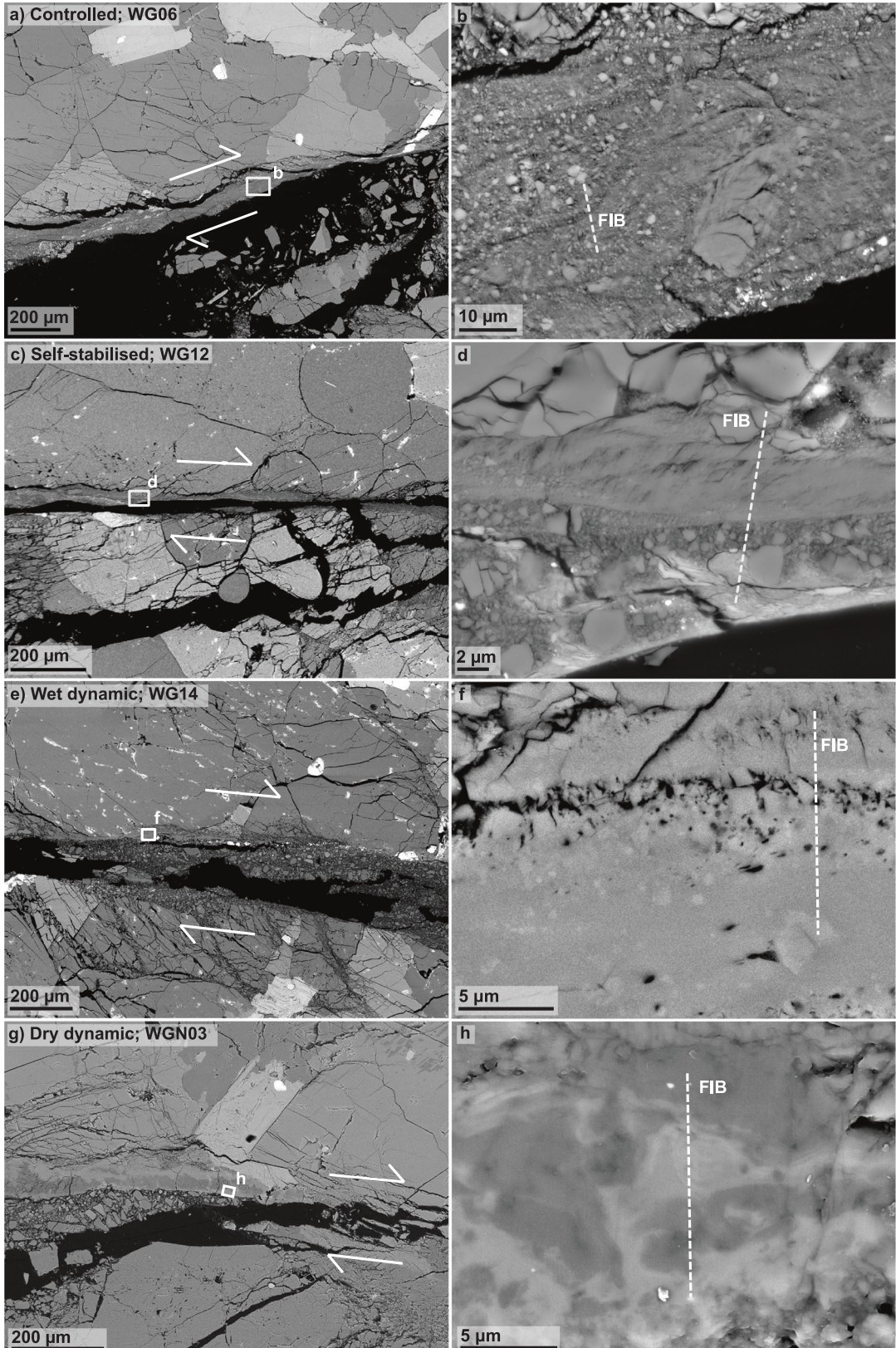

**Fig. 2 | Microstructures of the deformed samples.** Low (**a**, **c**, **e**, **g**) and high magnification (**b**, **d**, **f**, **h**) images taken in backscattered-electron (BSE) mode at the scanning electron microscope (SEM) revealing the fault gouges of the four different samples. Images **b**, **d**, **f**, and **h** highlight the locations for the focused-ion beam (FIB) cuttings to investigate the respective nanostructures at the transmission electron microscope (TEM).

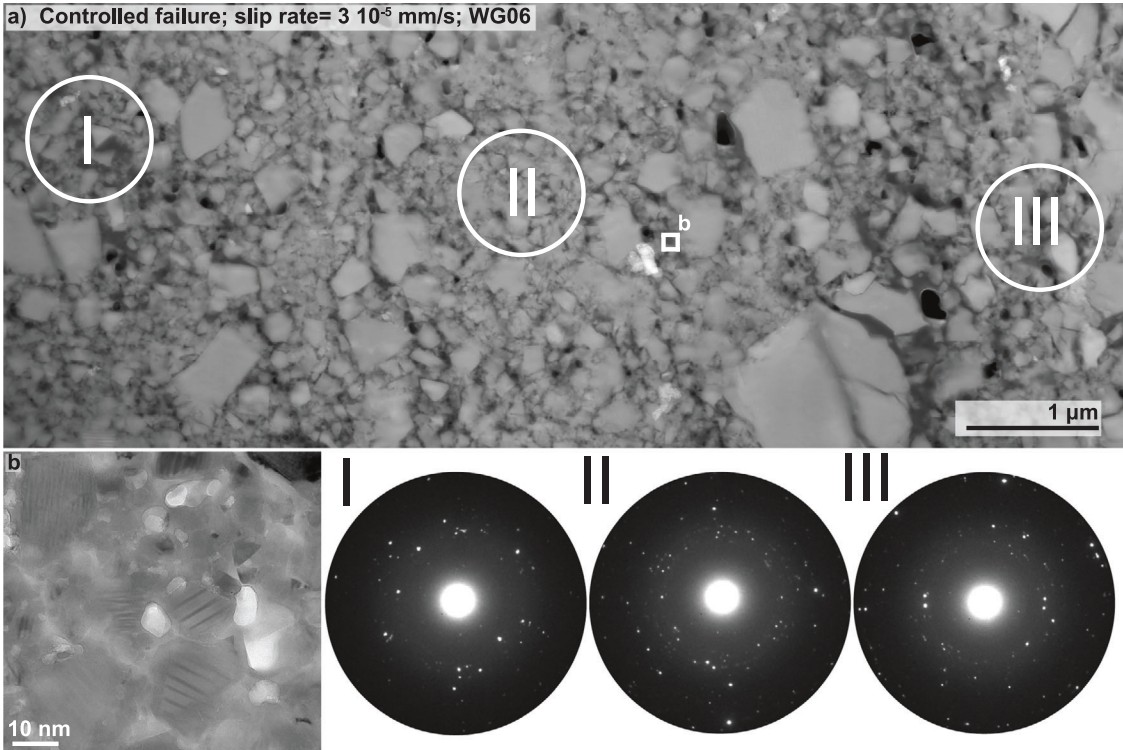

**Fig. 3 | Nanostructural analysis of sample WG06. a** Overview of the nanostructure of the focused-ion beam (FIB) section cut from sample WG06 that failed in a controlled manner (slip rate of around $3 \times 10^{-5}$ mm/s). The location of the FIB section is highlighted in Fig. 2b. The selected area electron diffraction (SAED) patterns I to III within this zone, almost exclusively made of feldspar, reveal numerous diffraction spots and a faint halo, which is most likely the result of the mainly sub-micron sized grains and even a significant amount of grains that are <10 nm in size (bright field (BF) image in (**b**), responsible for the diffuse halo.

recorded displacement ($\delta$) from the beginning of the test ($\delta = 0$) until its end ($\delta_{final}$). We calculated a total energy density ranging from around 170 kJ.m$^{-2}$ (controlled) to around 790 kJ.m$^{-2}$ (dry dynamic; Fig. 9). Dividing the total energy density by the average duration of the rupture process gives us a rough estimate on the power expended per unit area. We estimated a power of 5 W.m$^{-2}$ for the controlled run to $285 \times 10^{6}$ W.m$^{-2}$ for the dry dynamic test (Fig. 9). Power dissipation of the order of 10 to 100 MW.m$^{-2}$ is what is anticipated for dynamic slip at depth during natural earthquakes (see Fig. 4 in Di Toro et al.[31]).

Work expended per unit area obviously differs between the tests, however, the total energy density between the tests only changes by a factor of around 3–5 whereas power varies by 8 orders of magnitude. As gouge nanostructures are significantly different, it seems reasonable to state that such differences are best explained by changes in power rather than work.

**Underlying process for amorphization – comminution vs. melting**

Grain comminution and melting are both material-specific processes that can lead to the formation of an amorphous material. Yet, melting is a thermally activated process whereas comminution is insensitive to temperature changes. To determine whether the amorphous material, present to different degrees in every gouge, is the result of comminution or in fact of frictional melting, we searched for unequivocal chemical signatures or structural markers for melting.

In contrast to the controlled and the self-stabilised samples, both dynamic samples exhibit two strong structural markers indicative of frictional melting within the fault gouge. The first strong evidence is structural indicators for partial melting of biotite and orthoclase. Biotite grains that are truncated by the fault exhibit the formation of vesicles (Fig. S2g, h), and biotite remnants, located within the fault gouge, demonstrate lenses, probably filled with quenched melt, which

are strikingly similar to partially molten biotite grains (Figs. 8b and S3f; see Fig. 3a in ref. 48). Structures indicating partial melting of orthoclase are lens-shaped features within a large orthoclase grain (Fig. 6), as well as sutured or wavy orthoclase grain boundaries (Figs. S2d and S3d). The lenses within the orthoclase grain all exhibit a uniform orientation along (010)–feldspar's perfect cleavage and albite twin plane–and are filled with material that, judging by the similarity in brightness contrast, has the same or a very similar chemical composition relative to the amorphous material surrounding orthoclase. Melting is a process that takes place along interfaces, such as grain boundaries, cleavage planes, or other defects, and the lens-shaped features are another example for intracrystalline melting in feldspar previously observed in several other studies (e.g. refs. 49,50). Furthermore, we observe that the respective melt-lens thickness decreases with an increasing distance to the surrounding amorphous material. Assuming that the latter is in fact a quenched melt–a glass–the decrease in lens thickness fits the expected temperature gradient across the melt-fragment interface towards the fragment's centre (at least in 2D). As orthoclase has a higher melting point than biotite (around 650 ˚C), the observed partial melting of orthoclase requires a temperature rise on the fault beyond 1100 ˚C during rapid shearing (see Fig. 5 in Spray[51]).

The second strong evidence for melting is the occurrence of magnetite nanograins, which is restricted to fault gouges that formed in dynamically failing samples. Magnetite is a breakdown product of biotite dehydration at elevated temperatures of around 650 ˚C[48], and was already present as accessory phase in the starting material. Hence, these magnetite nanograins could be fragments. Yet, all nanograins show a very similar grain size and are euhedral in shape (Figs. 7b, S2f and S3e). Most importantly, magnetite nanograins exhibit concave grain boundaries strongly resembling so called Hopper crystals (Fig. 8c–g). Hopper growth occurs when the crystal edges grow considerably faster than the centres of the crystal faces. Such a crystal

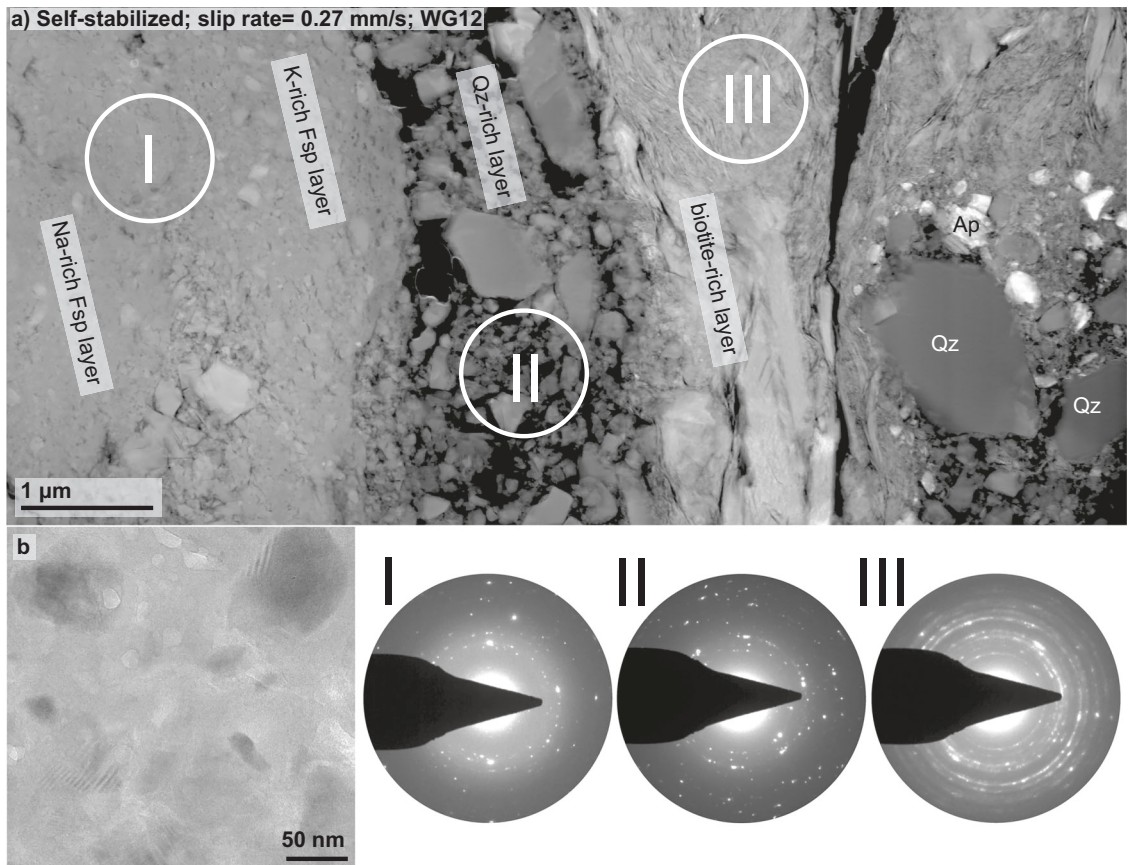

**Fig. 4 | Nanostructural analysis of sample WG12. a** Overview of the nanostructure of sample WG12 that failed in a self-stabilised mode (peak slip rate= 0.27 mm/s). The location of this focused-ion beam (FIB) section, presented in (**a**) is highlighted in Fig. 2d. **b** High magnification image taken in bright field (BF) mode of a Feldspar-rich layer showing that most of the material present appears amorphous (selected area electron diffraction (SAED) pattern I). The other SAED pattern of other phases present, reveal that each phase shows a different range in grain size and shape. Qz quartz, Ap apatite, Fsp feldspar.

shape is unlikely the result of comminution. We are thus confident that the magnetite nanograins are newly formed and the results of biotite breakdown. Furthermore, these nanograins are surrounded by a rim showing depletion in compatible elements in magnetite's structure, i.e. Fe, O, and Ti, and enrichment in elements that are incompatible, i.e. Si (Figs. 7c and 8c–g), reflecting that all elements were, at least temporarily, mobile to diffuse either out or into the nucleating and growing magnetite nanograins.

Given the lack of these structural indicators for melting in the micro- and nanostructures of the controlled and self-stabilised sample, respectively, we therefore state that the underlying mechanism for amorphization in the controlled and the self-stabilised failure samples is grain comminution and that the amorphous material in the dynamically failing samples is in fact glass produced through fast frictional melting, i.e. shear melting, and subsequent quenching. It seems likely that shear melting was preceded by grain comminution[52].

### Influence of fault gouge evolution on deformation dynamics

As we have established that slip rates indeed influence fault gouge structure, it is now of interest to assess whether differences in fault gouge structure impact fault dynamics during failure and thus to identify a potential feedback between fault dynamics and fault gouge structure. Specifically, the presence of sub-micron sized gouge or amorphous materials, including a quenched melt, could cause strong dynamic weakening, as independently reported in high velocity friction experiments with large slip distances (e.g. refs. 14,15,19,53). Thus, a strong feedback between fault dynamics and fault gouge structure is expected if the gouge contains sub-micron sized (e.g.

refs. 13,14,16–18,54,55 and references therein) or amorphous material as a result of comminution and/or frictional melting (e.g. refs. 19,20,22,31,56,57, and references therein).

The first step to compare the mechanical behaviour between different experiments is to separate the fluid pressure effect from other additional weakening (or strengthening) effects that could arise from intrinsic material properties. Assuming that shear strength is directly proportional to Terzaghi's effective normal stress, we can compute an apparent friction coefficient

$$f_{app} = \tau/(\sigma_n - P_f), \qquad (2)$$

which should reflect the intrinsic material shear strength. In Eq. (2), $\tau$ and $\sigma_n$ are the shear stress and normal stress on the fault, calculated from the differential and confining pressure data, and $P_f$ is the local fluid pressure measured on the fault during slip[46].

In the controlled experiment, we observe gradual weakening from the peak stress down to a constant residual, equivalent to a friction coefficient of 0.89 (Fig. 1), which is well within expectations for Westerly granite at 40 MPa effective stress[58]. In the self-stabilised experiment, the apparent friction is essentially indistinguishable from that of the dry test, indicating that no significant additional weakening occurred during slip at moderate slip rate compared to the controlled case (Fig. 1c). In the water-saturated experiment that resulted in dynamic rupture, the initial evolution in apparent friction also closely matches that of the other tests. As the apparent friction subtracts the effect of pore fluids on friction, this value can thus be regarded as a measure of the intrinsic frictional strength of a sample. At the onset of

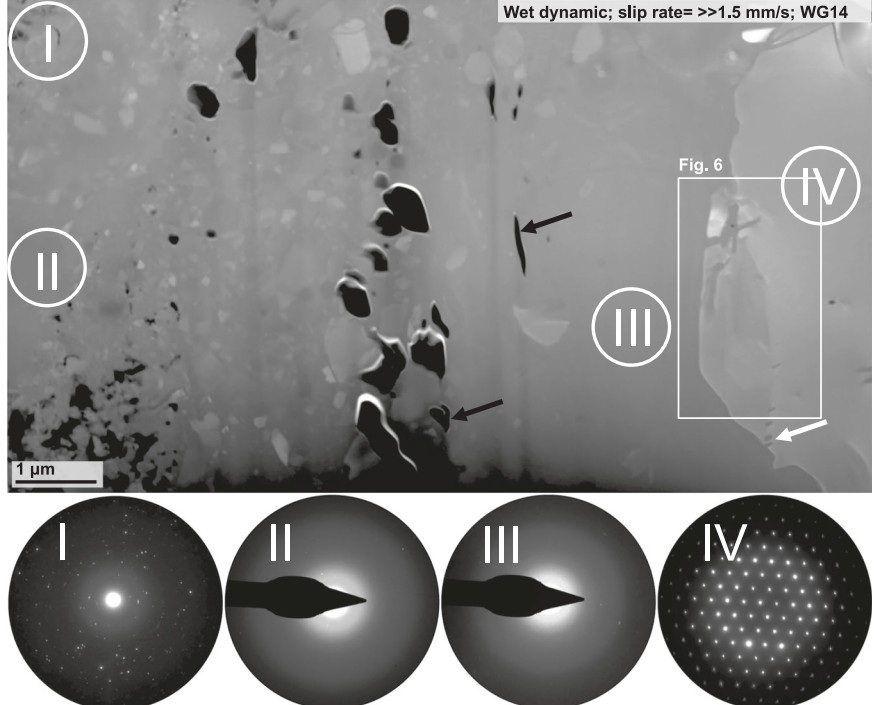

**Fig. 5 | Nanostructural analysis of sample WG14.** The image was taken in high-angle-annular dark-field (HAADF) mode at the transmission electron microscope (TEM). Several grains with various grains sizes can be found in a material that show no sign of crystallinity (selected area electron diffraction (SAED) pattern II and III). The large grain on the right-hand side was identified as orthoclase. Vesicles that show complex shapes, from more spherical to lense-shaped, are highlighted by black arrows.

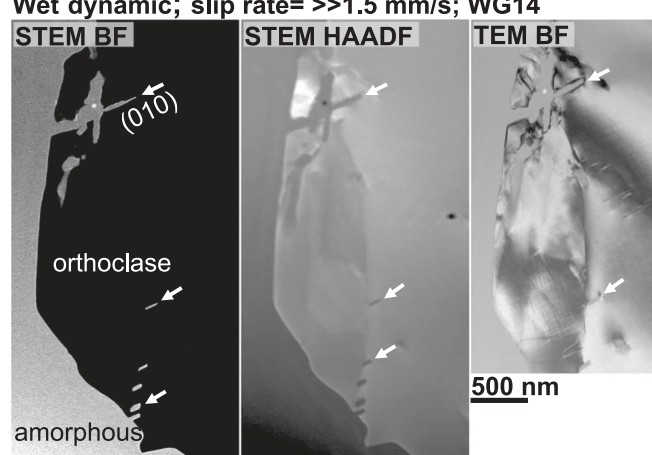

**Fig. 6 | Preferential melting along intracrystalline interfaces.** High magnification images taken in different modes at the transmission electron microscope (TEM) of a large orthoclase grain. White arrows mark lenticular zones that show the same brightness contrast as the amorphous material surrounding the orthoclase grain. STEM scanning transmission electron microscopy, BF bright field, HAADF high-angle-annular dark-field, TEM transmission electron microscopy.

dynamic slip and beyond, the stress data do not represent the actual strength of the fault, because inertial effects are not negligible. In particular, the piston's inertia might cause a stress overshoot, i.e. unloading below the material strength (e.g. ref. 59). Thus, the large drop in stress (and thus in apparent friction) cannot be directly associated with intrinsic weakening. Assessing the effect of piston inertia on the observed stress drop demonstrates that the large stress drop in the dynamic failure tests are not necessarily the manifestation of an

additional weakening processes, i.e. fault lubrication due to gouge melting in the dynamic samples (see the supplementary information for details).

Despite the uncertainty linking the fault dynamics with gouge melting in the dynamic samples, the fault gouge of the self-stabilised and the controlled samples both exhibit gouges filled with sub-micron sized and even amorphous material produced by extreme comminution, and yet, we observed no influence on fault dynamics, which is in agreement with the results of Yund et al.[27]. Since fault lubrication due to sub-micron sized or amorphous gouge is linked to viscous deformation of the gouge material (e.g. refs. 13,14,16–18,54,55, and references therein), we assume that the absence of fault lubrication in our samples is caused by the difficulty to activate plastic deformation mechanisms in feldspar relative to other rock-forming minerals such as calcite and even quartz (e.g. refs. 60,61), especially within such short durations of dynamic slip of less than a second. As feldspar minerals represent the most abundant mineral group in crustal rocks, our observation that nanocrystalline or amorphous feldspar gouge, produced by mechanical grinding (samples WG06 and WG12; Figs. 1, 3, and 4) has no lubricating effect, appears relevant to better understand fault dynamics in silicate host rocks of the upper crust.

## Comparison with previous tests on granitoid samples

Many experiments have been conducted on granitoid gouge and especially on sawcut samples of Westerly granite. Yet, only a fraction of these studies has investigated the generated fault gouges of the recovered samples in more detail; In particular, microstructural analyses of initially intact samples are scarce.

In a relatively recent study by Lockner et al.[22] on sawcut samples, the authors document dynamic weakening caused by the formation of a continuous melt film that formed on the fault gouge during rapid shearing (see Figures in Moore et al.[56]. Fault gouges in our dynamic samples appear identical to their sawcut counterparts, which

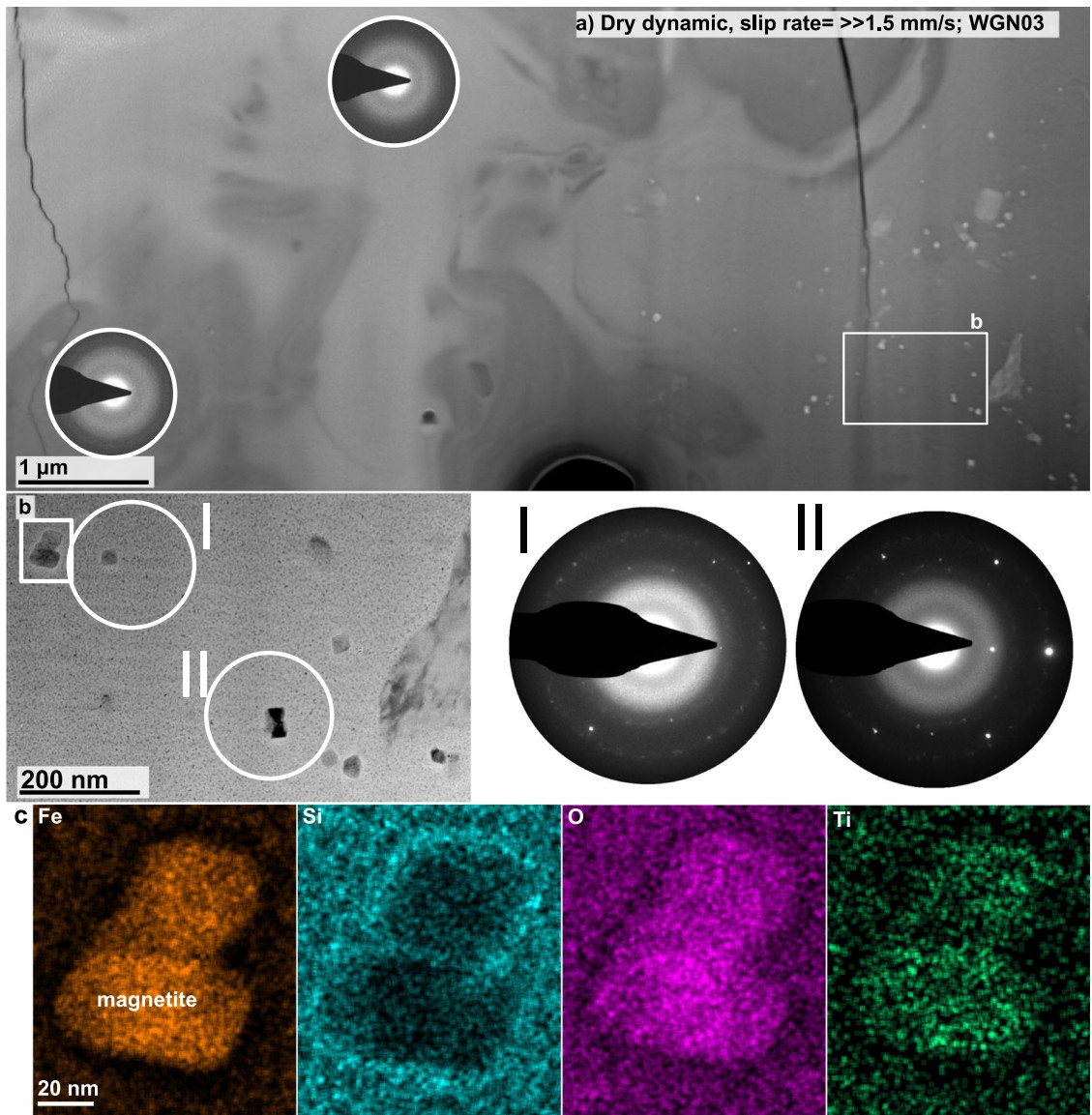

**Fig. 7 | Nanostructures of sample WGN03. a** Overview image, taken in high-angle-annular dark-field (HAADF) mode, showing the nanostructure of the first focused-ion beam (FIB) section cut from sample WGN03 (dry dynamic; peak slip rate ≫ 1.5 mm/s; Fig. 2h). The white rectangle marks the location of the image shown in **b**, which was taken in bright field (BF) mode. There is little indication that the fault gouge material is crystalline. Numerous euhedral crystals, appearing brighter in HAADF mode (**a**) or darker in BF mode (**b**) than the remaining material, were identified as magnetite using the selected area electron diffraction (SAED) pattern **II** together with the chemical information obtained through element distribution maps (**c**).

experienced a similar total slip of just a few mm[22,56]. We find this observation quite remarkable as it implies that changes in grain size and fault roughness are most pronounced during the very early stages of slip, i.e. within the first $\mu$m to mm of slip. Despite the striking similarities between the microstructures reported by Lockner et al.[22] and Moore et al.[56] with our microstructures (Figs. 2 and S2), the large stress drops observed here are found to be explainable by piston inertia and do not necessarily require enhanced dynamic weakening (see supplementary information). Reasons for this discrepancy are manifold. It is possible that we simply cannot resolve such a dynamic weakening effect. Let us however assume that the observed discrepancy is not the result of technical differences but has a physical root. As the main difference between Lockner et al.[22] and the present study is the use of sawcuts vs. initially intact samples, we believe that differences in fault roughness may resolve this discrepancy. Initially intact samples show a higher roughness than artificially produced fault surfaces of sawcuts, which may influence the melt's connectivity[19,20] – especially after low displacements, which is the case for the present

study and for Lockner et al.[22]. Such a relation between the initial state of the system and fault dynamics could highlight that in some situations, e.g. intact rocks or healed/sealed faults, larger displacements are required to result in dynamic weakening than indicated from tests on sawcut samples (e.g. ref. 31, and references therein).

Several studies, conducted under much higher confining pressures and shear stresses on chemically similar material to Westerly granite, observe micro- and nanostructures strikingly similar to our fault gouges, which developed during controlled and self-stabilised failure[33–35,38]. Marti et al.[35] concluded that there should be a stress-strain trade-off for amorphization as they observe amorphization after shear strains $\gamma$ of 10-20 and shear stresses of around 1.2 GPa whereas Yund et al.[27] document amophization of Westerly granite after deformation to large shear strains of $\gamma$ = 100-1000 under low stress conditions of just a few tens of MPa. Such a stress-strain trade-off would fit the theoretical assumption that the controlling factor for grain crushing or amorphization is related to the work expended per unit area of a fault surface (see Eq. (1)). We provided estimates that clearly demonstrate

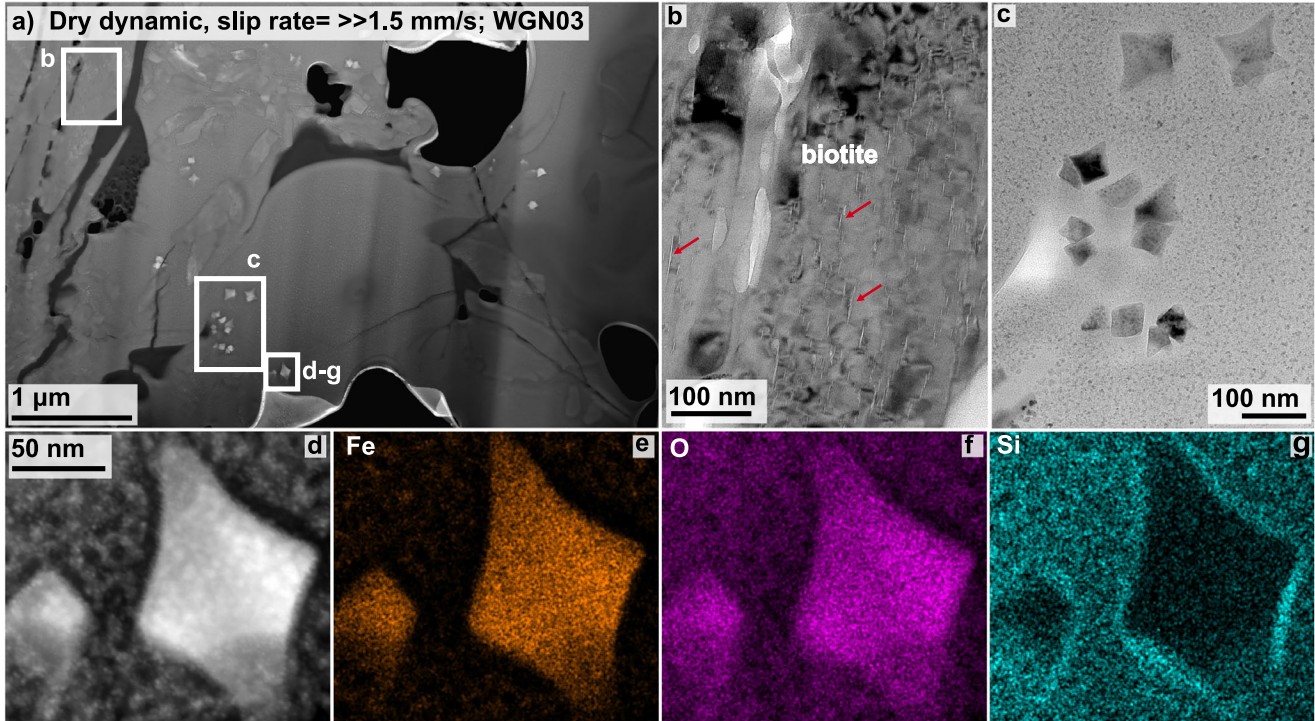

**Fig. 8 | Additional nanostructures of sample WGN03. a** Overview of the nanostructure revealed by the second focused-ion beam (FIB) section cut from sample WGN03 that experienced dynamic rupture, i.e. slip rates of ≫ 1.5 mm/s, in the absence of a pore fluid. We observe the breakdown of biotite highlighted by the formation of small lense-shaped features (red arrows in **b**) and the growth of nanocrystalline magnetite crystals (**c**–**g**). Many magnetite crystals show Hopper growth, i.e. concave shaped crystal edges, and depletion/enrichment halos around the crystals.

that it is power rather than total work controlling the structural evolution of the fault gouge (Fig. 9), in agreement with previous studies[31,62].

Beyond a critical slip rate, melting of the fault gouge material occurs, which is in accordance with previous tests on sawcut Westerly granite[63,64]. Interestingly, Stesky et al.[43] and Stesky[44] as well as Tullis and Yund[45], who all started with initially intact granite samples, did not find any evidence for melting—even after stick-slip experiments[43]. It is possible that evidence for melting was overlooked as the sampling volume for transmission electron microscope (TEM) is small and that the latter improved as a technique since the late 1970's. Furthermore, in our samples, glass-bearing parts of the fault gouge are often located at the interface between the fault gouge core and the adjacent wall rock, and thus not necessarily the most obvious location to search for glass-bearing fault gouge material.

## Link to nature

A goal of the present study was to find more resistant indicators for fast (seismic) slip rates than the occurrence of amorphous material. Magnetite nanograins are present in both dynamically failed samples and have been reported in glasses produced in Westerly granite during high velocity friction test[65], and have been found in natural pseudotachylites as well (e.g. ref. 66). A recent study by Papa et al.[67] exhibits microstructures of natural pseudotachylites found in granulite-facies paragneisses of the Serre Massif in Calabria (Southern Italy). Within the glassy pseudotachylite matrix, they report hercynite crystals of around 10 to 20 μm in diameter with concave crystal faces, indicating Hopper growth (see Fig. 4c, d in Papa et al.[67]). Since the experimental nanograins must have formed within less than a second to no longer than a second, as temperature drops rapidly after slip ceases, oxide micrograins found in natural high-grade metamorphic pseudotachylites could indicate rapid formation during coseismic slip and further

growth during the postseismic phase as host rock temperatures are high, i.e. granulite facies. The presence of oxide micro- or nanograins in natural pseudotachylites, even in high-grade metamorphic rocks, highlight that they can survive later recrystallisation and overprint. We therefore stress that oxide nanograins are in fact strong earthquake-indicators in natural rocks[6], and that—in the case of magnetite nanograins—will affect the magnetic properties across pseudotachylite-bearing faults[65]. It is important to note that the structural context in which magnetite occurs in granite matters. Here, iron oxide nanograins are clearly associated with the faulting event, but they can either have magmatic origins or be the product of fluid-rock interactions[68].

Moreover, our results demonstrate that gouge structures and textures normally associated with large displacements and/or elevated pressure and temperature conditions are already generated under conditions of the upper crust after minute amounts of slip, which matches observations in natural faults[69]. The reason we already observe zones of frictional melting in samples that dynamically failed but slipped only a few mm could be linked to the observation that, compared to other silicates such as quartz or calcite, feldspar readily amorphizes[27]. As amorphous material is highly reactive, melting could be facilitated, but also chemical reactions may be accelerated resulting in healing or sealing of the fault gouge (e.g. ref. 40), which could lead to a full restoring of the rocks original strength (e.g. ref. 15).

Much like in natural fault systems, we observe a complex localisation of strain within several zones across scales. At the micron scale, we observe that some vesicles appear to be highly stretched whereas others seem relatively unaffected by deformation (Figs. 2f and 5), demonstrating that strain can be highly heterogeneous even at the sub-micron scale.

Overall, our results clearly demonstrate a direct influence of slip rate on the syndeformational evolution of fault gouges. We observe a positive correlation and a causal relation between slip rates and the

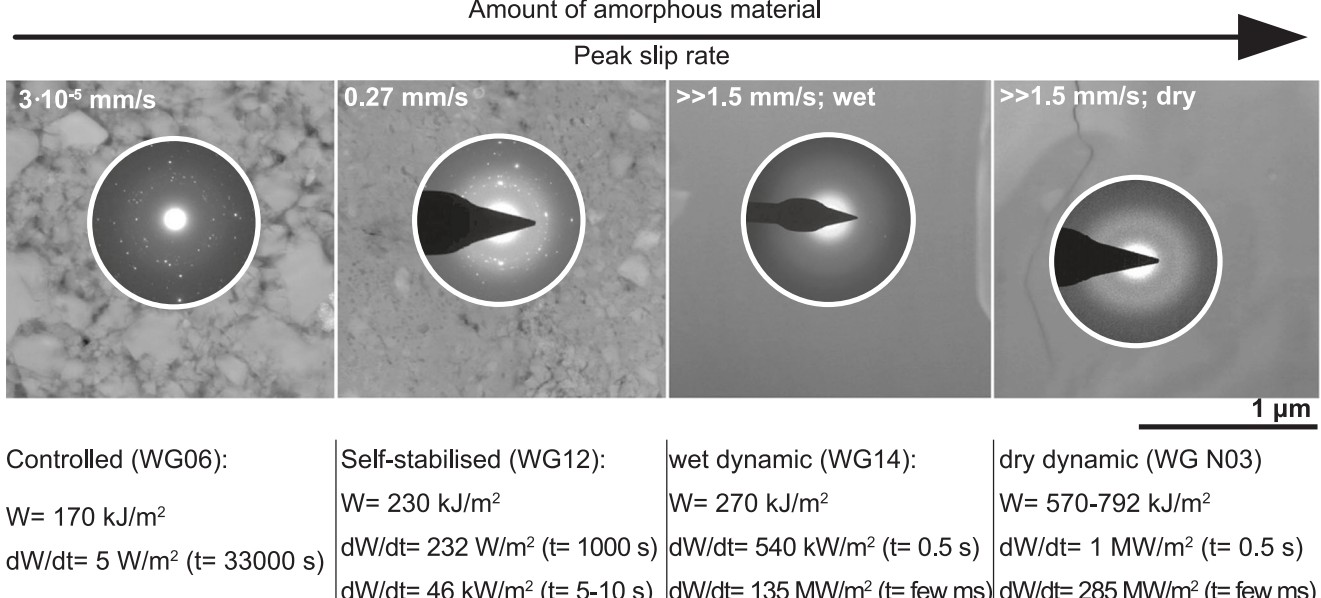

**Amount of amorphous material**

**Peak slip rate**

| 3·10⁻⁵ mm/s | 0.27 mm/s | >>1.5 mm/s; wet | >>1.5 mm/s; dry |

1 µm

Controlled (WG06):

W= 170 kJ/m²

dW/dt= 5 W/m² (t= 33000 s)

Self-stabilised (WG12):

W= 230 kJ/m²

dW/dt= 232 W/m² (t= 1000 s)

dW/dt= 46 kW/m² (t= 5-10 s)

wet dynamic (WG14):

W= 270 kJ/m²

dW/dt= 540 kW/m² (t= 0.5 s)

dW/dt= 135 MW/m² (t= few ms)

dry dynamic (WG N03)

W= 570-792 kJ/m²

dW/dt= 1 MW/m² (t= 0.5 s)

dW/dt= 285 MW/m² (t= few ms)

**Fig. 9 | Conceptual sketch exhibiting the observed positive correlation between the amount of amorphous material found in fault gouges with peak slip rate.** To qualitatively assess the degree of amorphization, the spot size for selected area electron diffraction (SAED) was kept constant. As comminution and melting are material specific processes, we selected areas that are exclusively made of feldspar. The calculated work ($W$) and power ($dW/dt$) together with the measured or estimated failure durations ($t$) demonstrate that power changes by 8 orders of magnitude whereas work only by a factor of 3 to 5. The range of $W$ in the dry dynamic case results from either using the full shear stress - slip curve to compute $W_{tot}$ (upper bound estimate) or only the fraction of the work that corresponds to pre-peak slip and an idealised slip weakening behaviour from peak stress to a residual strength of 40 MPa with a slip weakening distance of 1.5 mm as in the quasistatic dry test.

degree in amorphization of the gouge material pointing towards power being the determining parameter controlling the fault gouge structure. Gouge melting, identified by the growth of magnetite (iron oxide) nanograins and partial melting of orthoclase, is restricted to samples that failed dynamically at slip rates of ≫1.5 mm/s. As iron oxide micrograins have been found in natural pseudotachylites, we highlight that such oxide grains could be robust indicators for seismic slip as they appear to survive later overprinting events. It remains unclear whether the observed stress drop in the dynamic samples is in fact linked to melt lubrication or a result of piston inertia. However, it is obvious that the abundant sub-micron sized and even amorphous gouge material, found in the controlled and self-stabilised samples and formed due to mechanical grinding, had no weakening effect on fault strength in granite samples. As easy amorphization of feldspar could facilitate melting, even microseismic events (Mw < 2), could produce melt patches in granitoid rocks that can become ingredients of pseudotachylites.

## Methods

### Starting material and sample preparation

We used intact drill cores (4 × 10 cm) of Westerly granite, containing around 44% plagioclase, 28% quartz, 21% alkali feldspar (mostly orthoclase), 6% mica (mainly biotite and minor muscovite), and 1% accessory minerals, e.g. titanite, apatite, zircon, magnetite,[70]. Recovered samples were embedded in epoxy, placed under vacuum, and then cured at ambient conditions. Afterwards, samples were cut lengthwise to the fault. We produced two to three thin sections of one-half of each sample, which were then carbon coated to be investigated at the electron microscopes.

### Experimental methods

Samples were thermally cracked by placing them into a tube furnace and exposing them to 600 °C for 2 h. The heating rate was set to 3 °C.min⁻¹ and cooling was initialised by switching off the furnace. It took roughly 12 h until ambient temperature was reached. Except for sample WGN03, all drill cores were initially notched by cutting two 17 mm deep notches at an angle of 30° at opposite sides into the top and bottom part of the cylinders, respectively. These notches were filled with Teflon disks, which are expected to only control the spatial formation of the fault during axial loading without influencing the sample's mechanical behaviour. Afterwards, samples were jacketed in a nitrile sleeve and placed in the triaxial apparatus, installed in the Rock and Ice Physics Laboratory at UCL. For dry tests, confining pressure was raised to the target pressure of 40 MPa (samples WG06 and WGN03). For water-saturated tests, confining pressure was initially raised to 20 MPa, then the samples were saturated by flowing through the pore fluid with a 5 MPa pressure difference, venting the downstream end to the atmosphere. After full saturation, confining and pore pressure were both raised to their target values, $P_c$, $P_f$ = 60, 20 MPa (sample WG14) or $P_c$, $P_f$ = 110, 70 MPa (sample WG12).

Samples WGN03 (dry), WG14 and WG12 (water-saturated) were deformed at constant axial deformation rate of 10⁻⁶/s, and deformation was stopped immediately after failure. Sample WG06 (dry) was initially deformed at 10⁻⁶/s until the peak stress was approached, and subsequent deformation was produced by carefully cycling the load, partially unloading the sample when accelerated deformation (as measured by acoustic emission activity) was detected.

### Analytical methods

Microstructural analyses were conducted using a Jeol JSM-6480LV SEM at University College London and a Zeiss Gemini 450 SEM at Utrecht University. Acceleration voltage and beam current on the sample were always set to 15 kV and 1 nA, respectively, for the acquisition of BSE images. To analyse the nanostructures of the recovered deformation samples, we cut FIB sections using a Helios Nanolab G3 FIB-SEM at Utrecht University. For the nanostructural investigation, we used a Talos F200X installed at Utrecht University—a TEM. All TEM analyses were performed using an acceleration voltage of 200 kV. Furthermore,

to qualitatively assess the fault gauges' crystallinity, we took SAED pattern. For the SAED pattern we kept the aperture size constant to enable comparison between samples. In most cases, we combined the structural information from SAED with chemical analyses, i.e. point measurements and/or element distribution maps, by using EDS.

## Data availability

The mechanical as well as the micro- and nanostructural data generated in this study are provided in the main text and in the supplementary information file.

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

## Acknowledgements

We are very grateful to Katharina (Tinka) Marquardt for discussing the observed nanostructures and their potential implications as well as to James Davy for his help with the sample preparation. This study received funding to S.I. from UK Natural Environment Research Council in the form of a NERC Independent Research Fellowship (Grant Agreement NE/W00805X/1), and this publication results from work carried out under Trans-National Access action under the support of EXCITE-EC-HORIZON 2020-INFRAIA 2020 Integrating Activities for Starting Communities under grant agreement N. 10100561 (EXCITE _C1_2022_18). Furthermore, this project has received funding to N.B. from the European Research Council (ERC) under the European Union's Horizon research and innovation programme (2020 project 804685/"RockDEaF" and 2024 project 101088963/"RockDeath"), the UK Natural Environment Research Council (Grant Agreement NE/M016471/1), and the Leverhulme Trust (Philip Leverhulme Prize).

## Author contributions

S.I. and N.B. conceptualised the study and acquired funding. F.A. and N.B. performed the deformation experiments. M.O. and S.I. conducted the micro- and nanostructural analyses. S.I. and N.B. wrote the first draft, and all authors (S.I., M.O., F.A., O.P., and N.B.) contributed to finalising the manuscript.

## Funding

## Competing interests

The authors declare no competing interests.
