## [Transparent Peer Review file · Nature Communications]

Nanostructures as indicator for deformation dynamics

Corresponding Author: Dr Sarah Incel

Version 0:

Reviewer comments:

Reviewer #1

(Remarks to the Author)

Dear Editor, dear authors,

This study systematically investigated the mechanical behavior of (notched and thermally cracked) Westerly granite under various deformation conditions, including dry and wet environments, as well as low and high applied pore pressures. The authors characterized the experimental products (the gouge layers) and subsequently determined the generation of amorphous materials (AMs) within the gouge layers, regardless of the experimental conditions. The authors suggested that the generation of melting-origin AMs does not correspond to the dynamic weakening, likely due to uncertainty from the effect of piston inertia. Therefore, the authors concluded that small seismic events can trigger the formation of AMs but may not influence frictional strength. I truly appreciate the efforts of the authors to conduct these studies.

In general, the methodology is sound, and the data support the current interpretation and conclusions. The manuscript is also well-written, though lacks some evidence and is sometimes written with imprecise scientific language (please see several minor comments). However, the finding of AMs formed after a slip seems not provocative (surprising), although the authors use intact rocks as the starting materials. The findings of nano-magnetite grains within the AMs are interesting, but its application (or implication) remains unclear. In addition, gouges formed during small slip events can be expected by fracturing and comminution processes (Mode II shear rupture propagating; Di Toro et al., 2009; Tao et al., 2023). At the current stage, I cannot tell the significant improvement of the work to the understanding in the related field.

I would suggest that the authors obtain some additional results and discuss the potential topics. For example, (1) Can the presence of AMs be relevant to fault (fracture) healing? Frictional melts can behave either weakening or strengthening during seismic slip (due to temperature variation and associated viscosity), and can heal the slip zone after slip. It seems that healing may be another approach if the authors perform slide-hold-slide experiments; (2) since magnetic properties are commonly used to estimate temperature during slip, what are the magnetic properties of nanomagnetite grains within the AMs; or (3) a comprehensive review of AMs. Integrated with the published studies on AMs, the authors can highlight that AMs can be formed under a wide range of deformation conditions on non-cohesive and cohesive rocks, and provide a broad application when fault zones contain AMs. These are things that haven't been explored and can provide insights into the field. Last but not least, the study documents frictional melting on gouges, references to gouge melting should be appropriate rather than rock melting. Some minor comments are made for this.

Minor comments

Line 27: It is not attractive (surprising), Please see the comments above.

Line 66: Han et al., 2014 GRL (Frictional melting of clayey gouge during seismic fault slip: Experimental observation and implications)

Line 67-69: Hung et al., 2019 JGR (Grain fragmentation and frictional melting During initial experimental deformation and implications for seismic slip at shallow depths) is more appropriate than Tsutsumi and Shimamoto, 1997.

Line 76-77: Nguyen et al., 2024 GRL (Fluid drainage leads to thermal decomposition of wet gouge during experimental seismic slip) for the case of water-saturated gouges.

Line 81-83: Mitchell et al. 2016 Geology (Fault welding by pseudotachylite formation) and Proctor and Lockner 2016 Geology (Pseudotachylite increases the post-slip strength of faults) can be the cases for change long-term strength of faults.

Line 142: 4.2 mm is an estimation instead of an experimental measurement.

Line 164: The so-called "flow textures" are not clear in the figures.

Line 169: cracks are cracks. "Cool" is an explanation (discussion).

Line 254-256: It seems risky to mention a positive correlation because the amount of AMs is calculated from the selected area. Are AMs distributed on both the cracked and notched surfaces?

Line 258-260: what does it mean?

Line 262-282: Does the calculation of work/power calibrate with the actual area of intact rocks (instead of the notched part)?

Line 310: viscosity is correlated to chemical composition and temperature (e.g., Giordano et al. 2008 (EPSL). Viscosity of magmatic liquids: a model. How do the authors know it is low-viscosity?

Line 336-342: I cannot understand this paragraph. Grain comminution is responsible for the AMs of the controlled and the self-stabilized failure samples. Melting is for the dynamic samples. Does grain comminution not take place in the dynamic samples? How?

Line 350: This discussion is about "fault gouge", so I recommend some papers on fault gouge that should be more appropriate than the ones currently quoted.

Line 357: Hung et al., 2019 JGR (Grain fragmentation and frictional melting During initial experimental deformation and implications for seismic slip at shallow depths) is more appropriate than Hirose and Shimamoto 2005 (cohesive gabbro rocks).

Line 363: Please remove Hirose and Shimamoto 2005 and Di Toro et al., 2006. Wu et al., 2020 JGR (Mixed-mode formation of amorphous materials in the creeping Zone of the Chihshang Fault, Taiwan, and implications for deformation style) and Wang et al., 2023 Geology (Melting of fault gouge at shallow depth during the 2008 MW 7.9 Wenchuan earthquake, China) should be appropriate.

Line 390-392: The statement is unclear, but Wang et al., 2023 Geology (Melting of fault gouge at shallow depth during the 2008 MW 7.9 Wenchuan earthquake, China) can be used for the statement.

Line 413: This paragraph can be made more solid with some discussion from the papers of gouge melting/AMs generation. Line 464-473: Is this paragraph necessary?

Line 475-498: The paragraph is about nanograins, which is not documented/highlighted in the previous paragraphs. It is a bit odd (to me) to jump out of his discussion.

Line 505-506: What kind of reactive property of AMs can facilitate melting?

Line 533-542: Please describe the details of the notched samples for readers. I have to check the sample preparation in the two papers cited.

Fig 4a: what is dark mica? Is it biotite?

Reviewer #2

(Remarks to the Author)

Review of "Nanostructures as indicator for deformation dynamics" by Incel et al.

This manuscript reported the fault gouge micro- to nanostructures produced by different slip histories during laboratory deformation experiments. Amorphization was observed in the gouge and the authors discussed how they were formed. The authors concluded that melting played a role for dynamic slip conditions whereas grain comminution produced amorphous materials. Lastly, the authors tried to connect the gouge texture with natural seismic events where amorphous materials can be produced even during microseismic events.

Overall, results are well-documented and discussions are thoroughly organized and supported by the presented results. In particular, "work vs. power" section seems to provide an important viewpoint to understand or interpret underlying physical process for fault slip, and supported by the experimental results as well as compilation of previous studies. I have several comments listed below, which I think are minor. I look forward to seeing this paper published soon in the journal.

Comments:

L204: Is there a possibility of selective melting or selective dissolution occurred in the case of self-established sample? In Figure 4, larger grains are remained in the Q-rich layer whereas only small grains are rich in Fsp layers, which seemed to me an evidence for feldspar grains being easier to become small (and eventually amorphous?) grains than quartz as documented in L505. If this observation is ubiquitous along the gouge, it would be better to be documented.

L249: SAED pattern can say the presence of amorphous materials but BF images cannot. For instance, Figure 4b contains some crystalline features (dark lines in a grain) and there are several bright spots in the SAED pattern. I totally agree with the presence of amorphous materials based on the halo in SAED patterns, but maybe mentioning BF images here to "confirm the presence of amorphous material" can be removed.

L280-282: How much power was required for grain comminution? For instance, is 5 W/m/m enough to induce grain comminution? Energy (and power) required for grain comminution can be roughly estimated by assuming grain size and surface energy. It would also be nice to provide a rough estimation of temperature rise at an asperity. How much temperature increase can be produced by the given slip velocity? Is it high enough to produce melt (>1100 C)? This consideration may also strengthen the discussion for the self-stabilized failure sample where grain comminution is thought to produce amorphous materials and flash heating temperature may be below the melting point.

L318: In Figure 6 the "lens-shaped features" appear to align along a boundary between darker and brighter regions in the HAADF image. Is the left part of orthoclase grain newly precipitated or crystallized during quenching? Or is it originally

presented there before melting? Readers (at least I) would like to know why the intracrystalline melting occurs because the presence of it may infer that temperature gradient is not only one reason for the control of melting process.

L406-407: "difficulty to activate plastic deformation" can also be induced by the quite short duration of dynamic slip where (bulk) temperature increase is not high enough to activate plastic deformation. Also, local depletion of water as monitored by local pore pressure drop also plays a role in deactivation of plastic deformation.

L411: I would suggest rephrasing "amorphous feldspar gouge has no lubricating effect" into "amorphous feldspar gouge has little lubricating effect for short, dynamic slip in intact rocks" because it would weaken if we consider a longer slip or slip on a preexisting fault (like Lockner et al., 2017) as the authors discussed later in the manuscript.

L507: I agree with the authors that melt patches can be formed even during microseismic events. However, I am a bit wondering if a "pseudotachylite" can be formed during a microseismic event as a "pseudotachylite" is often described as a continuous, straight melt in nature. This sentence may be rephrased something like: "even microseismic events ($M_w < 2$) could produce melt patches in granitoid rocks that can be ingredients of pseudotachylites."

Figure 3a: Please indicate shear direction. Same for Figures 4, 5, 7. Which side is pushed forward?

Figure 3b: Where is this close-up view taken from in panel a? Please indicate.

Figure 9: What did determine the range of "W" for the dry dynamic case? Please provide an explanation for this in the figure caption. Also, the unit for dW/dt maybe W/m^2 while it is W/m now in the figure.

Figure S2f: Please provide a scale bar in the panel.

Reviewer #3

(Remarks to the Author)

Review of "Nanostructures as indicator for deformation dynamics" by Incel et al.

The authors experimentally analyze the relationships between slip dynamics and gouge structure associated with experimental faults. The authors used intact granite samples that were loaded in a triaxial system under conditions of the upper crust with or without fluids. The samples failure by shear along the freshly formed faults occurred under quasi-static, and weak and full dynamic with displacement < 4 mm. The authors correlated the intensity of gouge amorphization with failure style. They found indication of melting and nanograins are common in samples deformed at high rates. They also correlated the nanostructure with power dissipation, energy input, and strength evolution. It is suggested that gouge nanotextures can form even during small slip events of intact rocks.

The authors presented a great piece of research which is professionally written and well organized. The manuscript can be published "as is." Admittedly, this opinion of mine (Ze'ev Reches) could be biased because the analysis agrees with results of research conducted by my group for years, as outlined below. The comments are divided into major comments (actually they are suggestions) and minor comments.

Ze'ev Reches

Major comments

1. First, my biased view. The current research approach is unique and important. Unlike most studies of earthquake stability, which analyze large displacement between two blocks at frictional contact, the authors started with a solid granite sample that fails and slips under continuous loading. This approach demonstrates the fundamental process of earthquake nucleation and rupture. Admittedly, I made this strong statement as it fits my view of earthquake instability. For example, Reches (1999) concluded: "The nucleation of earthquakes and the associated unstable slip propagation should be analyzed in terms of strength, unstable yielding and rupture of intact rocks." Muhuri et al (2003) stated: "...that gouge particles are cemented by chemical processes during hold periods and ...the cyclical strength variations are controlled by cohesion strengthening rather than by friction changes." Recently, Reches & Fineberg (2023) argued "...that earthquake dynamics are best understood in terms of dynamic fracture mechanics and not governed by the frictional properties of faults." (see: Reches, 1999, EPSL; Muhuri et al., 2003, Geology; Reches & Fineberg, 2023, JGR). This view is based on the ubiquitous observation that the cores of most fault zones are composed of fine-grained crushed rocks, and at typical earthquake depths (elevated temperature and presence of water, e.g., current experiments) such fine-grained material undergoes cementation and fast lithification. Slide-hold-slide experimental analyses showed that faults can regain their strength during period of hours to weeks (Karner et al., 1997; Muhuri et al., 2003). Similar healing strengthening was documented in-situ after major earthquakes (e.g. After Kobe earthquake in Tadokoro et al., 1999. S wave splitting...JGR 104, 981-991., and Li et al., 1998, Evidence of shallow fault zone strengthening after the 1992 M7.5 Landers, California, earthquake: Science, v. 279, p. 217–219).

2. The authors are aware of the possibility of fault zone cementation due to its fine grains (lines 58-61), but the main reason for using intact samples is gouge microstructure of a fresh fault (lines 97-100). While this is a good reason, as demonstrated

in the paper, it is suggested to discuss and emphasize the significance of the present approach for characterization of the microstructure associated with failure stability of a cemented, brittle intact fault zone. Fault cementation was studied experimentally with pre-cut samples (above), and the present work goes further and analyzes the effects of a "perfect & complete" lithification on microstructure and failure stability. The authors briefly mention healing and sealing (lines 91-94), but did not discuss the effect of these processes on fault cohesive strength, as it approaches the original strength (Fig. 4 in Reches and Lockner, 2010). I think that the possibilities of a fault to regain its cohesive strength should be discussed in the context of the present analysis.

3. The authors show (lines 27-30; 499-502) that their observations of amorphization and melting do not necessarily require large displacements and/or temperature. This is probably the most important conclusion of this study. Application of these experimental observations to field observations of small faults is not easy, probably due to reformation of the gouge material. In our analysis of faults in a syenite intrusion (Katz et al., 2003), we analyzed the microstructure of faults with displacements up to 20 mm. The fault zone is composed of fine-grain quartz, opaque oxides, and calcite (Fig. on left, Fig. 11 in Katz et al). While the general shape of this fault is similar to the present experimental faults, the original composition and sub-microstructure of the field fault were altered since its formation (line 59-61), so the slip stability cannot be inferred. See also the description of the gouge of Bosman fault in Wilson et al. (2006).

Minor comments (by line number)

117-125 the procedures of WG06 and WG12 are described too briefly, while understanding the slip rate control procedure of these samples in very central.

129-131 understood, but pity that this rate was measured...

145-148 this is not "friction coefficients", replace "shear/normal" ratio

169 , Fig 2g why are these 'cooling cracks'?

195-197 important observation

253-256 while this statement is apparently correct, there are only four samples here... so any "trend"

275-282 what are the relationships to previous studies? The 285 Mwm-2 is very high. And it is probably related to the energy dissipated in the extreme conditions at the rupture front (e.g., Reches and Dewers, 2005).

286-288 this is correct, yet note that comminution referred melting as it is easier to melt fine grains. Your line 342.

346-349 speculative...

384-395 This is a puzzling part. The piston is the loading system that reacts to stress changed of the sample, thus, unless stress measured ON the sample shows otherwise, one cannot separate the piston from the sample.

406 the experiments show "fault lubrication"

427-431 do you think that the weakening is due to piston inertia? I do not agree. The dynamic sample failure allows the piston to accelerate and not vice versa.

456-451 also Liao et al., 2014 (Fig.3)

Version 1:

Reviewer comments:

Reviewer #1

(Remarks to the Author)

Dear Editor, Dear authors

This is my second review of the manuscript. It is the revised version following the previous review. The authors did a good job responding accurately to most of the comments. Some issues were not addressed because I did not point them out specifically enough. Fortunately, the other reviewer raised similar comments, which improved the manuscript. The revision has significantly improved the quality of the paper, and it could serve as a guide for future work in this area. I have made several comments and recommend that the manuscript be considered for publication once these comments have been addressed.

Comments

Line 1-2. Title: The authors thoroughly document the production of amorphous materials (AMs) and their associated powder density. However, the current title is too general to convey this finding. I suggest highlighting this key finding in the title, as it

is highlighted in the manuscript (line 108-115). For example, "Frictional-Original Nanostructures as an Indicator for Deformation Dynamics" or "Gouge Amorphization as an Indicator for Deformation Dynamics." The authors can certainly do better than I can.

Line 259-264. Why is the intensity of the pure AMs different between WG14 and WG N03 (see Fig. 9)? Is it due to the different thicknesses of the FIB samples? Or, is it due to their different chemical compositions (biotite/orthoclase)? Could this affect the statement on powder density calculation?

Figure 4: The authors have confirmed that the dark mica is biotite, and it should be modified accordingly.

We are very grateful to the three reviewers for their effort and the time taken to provide feedback that, we find, improved our manuscript. All answers to the reviewers' questions are numbered and highlighted in red. Line references refer to the revised manuscript with track changes.

Reviewer #1 (Remarks to the Author):

Dear Editor, dear authors,

This study systematically investigated the mechanical behavior of (notched and thermally cracked) Westerly granite under various deformation conditions, including dry and wet environments, as well as low and high applied pore pressures. The authors characterized the experimental products (the gouge layers) and subsequently determined the generation of amorphous materials (AMs) within the gouge layers, regardless of the experimental conditions. The authors suggested that the generation of melting-origin AMs does not correspond to the dynamic weakening, likely due to uncertainty from the effect of piston inertia. Therefore, the authors concluded that small seismic events can trigger the formation of AMs but may not influence frictional strength. I truly appreciate the efforts of the authors to conduct these studies.

In general, the methodology is sound, and the data support the current interpretation and conclusions. The manuscript is also well-written, though lacks some evidence and is sometimes written with imprecise scientific language (please see several minor comments).

A1: Many thanks for your overall positive feedback on the manuscript structure and the data quality. It is however unclear to us where evidence is lacking as this is no further explained in the comments below, and stands somewhat in contrast to your evaluation that the interpretation is supported by the data provided. Furthermore, we are not sure in which part of the manuscript we used imprecise scientific language as precise line references or examples are missing.

However, the finding of AMs formed after a slip seems not provocative (surprising), although the authors use intact rocks as the starting materials.

A2: We, as well as reviewer #3, do not find it obvious at all to observe amorphous material in all fault gouges and even melting in the dynamically fractured samples that formed in initially intact rocks that slipped only a few mm. The existence of local amorphization and melting depends on the degree of strain localisation (and local strain/strain rate). Observing those textures and their consequences on slip dynamics in fault structures generated spontaneously (rather than forced, as in common friction tests on saw cuts/gouge) was an open problem. We believe that our observations are important for three main reasons. First, the presented micro- and nanostructures are identical to samples that already contain a fault (sawcut samples; Lockner et al., 2017, Passelègue et al., 2016) or are already gouge materials entirely that were exposed to large strains (e.g., Pec et al. 2012, 2012a, Marti et al. 2020). Consequently, this implies that – in granitoid rock – the formation of a “mature” fault gouge occurs in the early stages of slip (lines 464-467). Second, as we observe no dynamic weakening in the samples that underwent melting, we highlight that larger displacements are necessary than previously believed to cause dynamic weakening on faults that formed in initially intact or healed/sealed rocks as they are structurally more complex than straight sawcut samples (lines 479-487). Third, the only studies on initially intact granite samples, conducted several decades ago, observe no amorphous/glassy material within the gouge material (lines 506-515). Using state-of-the-art microscopes to analyse the micro- and nanostructures of the experimental fault gouges, we proved that amorphous material that formed by grinding or melting is present in fault gouges produced under similar experimental conditions. We added a few more sentences to the introduction to highlight the importance of

studying the potential coupling between fault dynamics and fault gouge structure to the introduction (lines 96-101).

The findings of nano-magnetite grains within the AMs are interesting, but its application (or implication) remains unclear. In addition, gouges formed during small slip events can be expected by fracturing and comminution processes (Mode II shear rupture propagating; Di Toro et al., 2009; Tao et al., 2023). At the current stage, I cannot tell the significant improvement of the work to the understanding in the related field.

A3: As stated in lines (352-369) we used the euhedral nanocrystalline magnetite grains as main proof that parts of the fault gouge material underwent melting. These nanograins are therefore absolutely crucial to differentiate between the origin of amorphous material – mechanical grinding vs. melting.

I would suggest that the authors obtain some additional results and discuss the potential topics. For example, (1) Can the presence of AMs be relevant to fault (fracture) healing? Frictional melts can behave either weakening or strengthening during seismic slip (due to temperature variation and associated viscosity), and can heal the slip zone after slip. It seems that healing may be another approach if the authors perform slide-hold-slide experiments; (2) since magnetic properties are commonly used to estimate temperature during slip, what are the magnetic properties of nanomagnetite grains within the AMs; or (3) a comprehensive review of AMs. Integrated with the published studies on AMs, the authors can highlight that AMs can be formed under a wide range of deformation conditions on non-cohesive and cohesive rocks, and provide a broad application when fault zones contain AMs. These are things that haven't been explored and can provide insights into the field. Last but not least, the study documents frictional melting on gouges, references to gouge melting should be appropriate rather than rock melting. Some minor comments are made for this.

A4: To (1) Understanding the coupling between amorphization and healing is, we believe, another research problem and thus beyond the scope of the present study.

To (2): Nakamura et al. (2002) studied the influence of the formation of magnetite nanograins due to melting of granite on the magnetic properties of the rock. We added an additional sentence to demonstrate that the growth of magnetite nanocrystals will indeed affect the magnetic properties of the rock (lines 537-538).

To (3): We agree that a thorough review of the ways amorphous material may or may not form in cohesive and non-cohesive rocks over a broad range in conditions would be indeed very interesting. However, we hope that the reviewer understands that this would be a separate study in itself.

Minor comments

Line 27: It is not attractive (surprising), Please see the comments above.

A5: We believe that demonstrating that gouge textures, which are generally associated with faults that experienced large displacements can also be found in faults that are devoid of large slip and formed in initially intact (cohesive) rocks is in fact rather surprising or at least not obvious. Please see our answer A2 above.

Line 66: Han et al., 2014 GRL (Frictional melting of clayey gouge during seismic fault slip: Experimental observation and implications)

A6: There is plenty of research conducted on gouge material. Due to the limitations in references, we decided to focus on granite gouges and only cite other studies that involved different material where necessary.

Line 67-69: Hung et al., 2019 JGR (Grain fragmentation and frictional melting During initial experimental deformation and implications for seismic slip at shallow depths) is more appropriate than Tsutsumi and Shimamoto, 1997.

A7: It is not obvious to us how the suggested publication is “more appropriate” than Tsutsumi & Shimamoto (1997): they were the first to ever evidence the “viscous break” effect that we mention in the text. Many other publications have followed that document this effect further, including Hirose and Shimamoto, JGR 2005, DelGaudio et al., JGR 2009; Niemeijer et al., JGR 2011, to name a few. That being said, the suggested study by Hung et al. (2019) is indeed relevant here and we now make a reference to it.

Line 76-77: Nguyen et al., 2024 GRL (Fluid drainage leads to thermal decomposition of wet gouge during experimental seismic slip) for the case of water-saturated gouges.

A8: We hope that the reviewer understands that due to the limitation in references, and the abundance of publications on fault gouge experiments, we decided to only cite studies that used similar materials, i.e., granitoid samples. In the suggested publication, the authors used kaolinite fault gouge, which is chemically quite far away from our starting material.

Line 81-83: Mitchell et al. 2016 Geology (Fault welding by pseudotachylyte formation) and Proctor and Lockner 2016 Geology (Pseudotachylyte increases the post-slip strength of faults) can be the cases for change long-term strength of faults.

A9: We are now referring to the publication by Mitchell et al. (2016; line 84).

Line 142: 4.2 mm is an estimation instead of an experimental measurement.

A10: We changed the sentence to: Total accumulated slip was estimated to range between 1.2 to 4.2 mm. (Lines 148-149).

Line 164: The so-called “flow textures” are not clear in the figures.

A11: Agreed. As flow textures can be highly subjective, they should therefore not be used to infer fault dynamics, which is a point we now make in line 80.

Line 169: cracks are cracks. “Cool” is an explanation (discussion).

A12: We agree that cracks are cracks. However, cooling cracks is an established term and stating that these cracks resemble cooling cracks is, we believe, not yet an interpretation but a rather a description. However, we understand that referring to these structures as cooling cracks seems confusing, which is why we are now no longer describing these structures as cooling cracks.

Line 254-256: It seems risky to mention a positive correlation because the amount of AMs is calculated from the selected area. Are AMs distributed on both the cracked and notched surfaces?

A13: We only investigated the fault gouge material that was produced during fracture and slip of the unnotched parts of the sample. As only selected area electron diffraction (SAED) provides robust evidence for the presence of amorphous material (Fig. 9), which can only be obtained through nanostructural analyses, we are restricted to probing a tiny fraction of the samples’ fault gouge materials. However, to render this comparison less ‘risky’ we decided to select the most representative zones identified using backscatter electron and only compare a phase that is most abundant, i.e., feldspar (lines 194-200).

Line 258-260: what does it mean?

A14: We deleted this sentence, because it is redundant (lines 264-268).

Line 262-282: Does the calculation of work/power calibrate with the actual area of intact rocks (instead of the notched part)?

A15: To avoid taking into account the absolute area that slipped, we calculated the total work per unit area, as described in lines 270-273. Therefore, the actual area does not enter the calculation.

Line 310: viscosity is correlated to chemical composition and temperature (e.g., Giordano et al. 2008 (EPSL). Viscosity of magmatic liquids: a model. How do the authors know it is low-viscosity?

A16: It is correct that we do not know the exact viscosity of the frictionally produced melt, and we understand that discussing the influence of viscosity (even qualitatively) on the formation of new crystallites is perhaps confusing. We re-arranged the subsection Underlying process for amorphization – comminution vs. melting and deleted this part (lines 320-323).

Line 336-342: I cannot understand this paragraph. Grain comminution is responsible for the AMs of the controlled and the self-stabilized failure samples. Melting is for the dynamic samples. Does grain comminution not take place in the dynamic samples? How?

A17: We believe that grain comminution preceded melting in the dynamic samples as written in lines 379-380.

Line 350: This discussion is about "fault gouge", so I recommend some papers on fault gouge that should be more appropriate than the ones currently quoted.

A18: Here we used the term fault gouge in a general sense, not specifically related to "gouge experiments": even in initially bare rock surface tests, gouge spontaneously forms due to asperity wear, so experimental work on bare-rock surfaces is also relevant. This is especially the case here where we focus part of our discussion on the process frictional melting, which has been extensively documented in (initially) rock-on-rock friction tests. In this section we also make reference to "gouge experiments" where appropriate, keeping in mind that we aim to discuss processes and not specific experimental conditions.

Line 357: Hung et al., 2019 JGR (Grain fragmentation and frictional melting During initial experimental deformation and implications for seismic slip at shallow depths) is more appropriate than Hirose and Shimamoto 2005 (cohesive gabbro rocks).

A19: The paper by Huang et al. is also on bare rock surfaces; it is certainly relevant to our discussion, and we now cite it here (line 69).

Line 363: Please remove Hirose and Shimamoto 2005 and Di Toro et al., 2006. Wu et al., 2020 JGR (Mixed-mode formation of amorphous materials in the creeping Zone of the Chihshang Fault, Taiwan, and implications for deformation style) and Wang et al., 2023 Geology (Melting of fault gouge at shallow depth during the 2008 MW 7.9 Wenchuan earthquake, China) should be appropriate.

A20: The papers by Wu et al. and Wang et al. are certainly appropriate, but due to limitations in references, we now cite Wang et al. (2023) as the fault gouge material described in this paper, i.e., granitoid gouge material, is similar to the starting material selected for the present study (line 402).

Line 390-392: The statement is unclear, but Wang et al., 2023 Geology (Melting of fault gouge at shallow depth during the 2008 MW 7.9 Wenchuan earthquake, China) can be used for the statement.

A21: We are not quite sure what exactly is unclear in our explanation: we are discussing the role of piston inertia, which can lead to a discrepancy between the recorded stress and fault strength. This is an experimental artifact that prevents us from drawing definite conclusions regarding the strength of our lab faults during dynamic slip. If we understand correctly, the paper by Wang et al. is a field and lab report on pseudotachylytes in a natural fault material, but it does not seem relevant to this discussion.

Line 413: This paragraph can be made more solid with some discussion from the papers of gouge melting/AMs generation.

A22: In this section, we tried our best to discuss the formation process (specifically, the conditions in terms of strain, strain rate, degree of localisation, pressure, temperature) of the textures observed in our samples, and see how clearly they can be evidence for seismic slip. We are thus focused mainly on comparing our observations to those on similar rock types in different conditions, extending references to key papers on high velocity friction when relevant. To this end, we cite publications that discuss fault gouge amorphization and melting, e.g., Lockner et al. (2017) and DiToro et al., (2006, 2011) or the lack thereof, e.g., Tullis and Yund (1977) and Stesky et al. (1974). The extended discussion of the work of Lockner et al. 2017 (and Moore et al., 2016) is thus directly relevant to the topic, since they used the exact same starting material in triaxial conditions, but on saw-cut samples, and document textures from small slip events at high stress conditions (similar to ours). We are not quite sure where the reviewer thinks the section needs to be more solid and which specific reference would be missing to help our discussion.

Line 464-473: Is this paragraph necessary?

A23: Since we cite the only studies on intact westerly granite that – as far as we know—exist, we believe that it is crucial to discuss any similarities or difference between the present and these previous studies.

Line 475-498: The paragraph is about nanograins, which is not documented/highlighted in the previous paragraphs. It is a bit odd (to me) to jump out of his discussion.

A24: The magnetite nanograins are extensively documented in lines 183-191 and 243-250. The overall aim of the paper is not to document amorphization and melting in general, but to specifically determine key microstructural indicators of slip dynamics. Magnetite nanograins are such an indicator, relevant to field studies.

Line 505-506: What kind of reactive property of AMs can facilitate melting?

A25: As stated above in A17, we believe that grain comminution preceded melting and that melting was probably facilitated due to the higher surface area of amorphous material as already suggested by Spray (1987; lines 373-380).

Line 533-542: Please describe the details of the notched samples for readers. I have to check the sample preparation in the two papers cited.

A26: We added more details on the sample geometry in lines 596-605.

Fig 4a: what is dark mica? Is it biotite?

A27: The dark mica, we are referring to, is biotite.

Reviewer #2 (Remarks to the Author):

Review of “Nanostructures as indicator for deformation dynamics” by Incel et al.

This manuscript reported the fault gouge micro- to nanostructures produced by different slip histories during laboratory deformation experiments. Amorphization was observed in the gouge and the authors discussed how they were formed. The authors concluded that melting played a role for dynamic slip conditions whereas grain comminution produced amorphous materials. Lastly, the authors tried to connect the gouge texture with natural seismic events where amorphous materials can be produced even during microseismic events.

Overall, results are well-documented and discussions are thoroughly organized and supported by the presented results. In particular, “work vs. power” section seems to provide an important viewpoint to understand or interpret underlying physical process for fault slip, and supported by the experimental results as well as compilation of previous studies. I have

several comments listed below, which I think are minor. I look forward to seeing this paper published soon in the journal.

Comments:

L204: Is there a possibility of selective melting or selective dissolution occurred in the case of self-established sample? In Figure 4, larger grains are remained in the Q-rich layer whereas only small grains are rich in Fsp layers, which seemed to me an evidence for feldspar grains being easier to become small (and eventually amorphous?) grains than quartz as documented in L505. If this observation is ubiquitous along the gouge, it would be better to be documented.

A28: Since grain comminution is indeed a material-specific process (e.g., Yund et al., 1990), we only compare the nanostructures of feldspar-rich sites between the samples (lines 194-200) as feldspar is the most abundant mineral phase in Westerly granite. Because only nanostructural investigations reveal unambiguous results on the crystallinity of the material, we are unfortunately unable to probe the entire fault gouge.

L249: SAED pattern can say the presence of amorphous materials but BF images cannot. For instance, Figure 4b contains some crystalline features (dark lines in a grain) and there are several bright spots in the SAED pattern. I totally agree with the presence of amorphous materials based on the halo in SAED patterns, but maybe mentioning BF images here to “confirm the presence of amorphous material” can be removed.

A29: We agree that the SAED are crucial to assess whether a material is crystalline or not, and we understand the point raised about the BF images, but we decided to keep this statement. As a rule of thumb anything that is smaller than 10 nm is considered amorphous (e.g., Yund et al., 1990), as one can no longer speak of a long-range periodicity of the crystal lattice. We therefore decided to also show such BF images to demonstrate that there are numerous grains within the fault gouge of the samples that failed in a controlled manner that are smaller than 10 nm.

L280-282: How much power was required for grain comminution? For instance, is 5 W/m/m enough to induce grain comminution? Energy (and power) required for grain comminution can be roughly estimated by assuming grain size and surface energy. It would also be nice to provide a rough estimation of temperature rise at an asperity. How much temperature increase can be produced by the given slip velocity? Is it high enough to produce melt (>1100 C)? This consideration may also strengthen the discussion for the self-stabilized failure sample where grain comminution is thought to produce amorphous materials and flash heating temperature may be below the melting point.

A30: We agree that calculating the temperature rise can be helpful in situations where microstructural evidence is lacking. However, in the nanostructures of our samples we can directly observe which phases started to melt (biotite and orthoclase), providing us with a rather robust estimate for the temperature rise on the fault during frictional heating (lines 304-369). Since the calculation of temperature rise, e.g., at asperities, necessarily requires some assumptions, we believe that -- in our case -- using ‘real’ evidence from the nanostructure is much more reliable to estimate the temperature rise on the fault than providing calculations based on assumptions, e.g., on heat conductivity, local slip rate, asperity size etc.

L318: In Figure 6 the “lens-shaped features” appear to align along a boundary between darker and brighter regions in the HAADF image. Is the left part of orthoclase grain newly precipitated or crystallized during quenching? Or is it originally presented there before melting? Readers (at least I) would like to know why the intracrystalline melting occurs because the presence of it may infer that temperature gradient is not only one reason for the control of melting process.

A31: This is a very good observation, and we thank the reviewer for pointing this out. We spent a significant amount of time trying to analyse this orthoclase fragment. We also thought about the possibility that the left part of this fragment, as pointed out by the reviewer, could have been a newly precipitated or re-crystallised region. We, however, did not find any evidence for the latter. In contrast, since the melt patches are all oriented in the same direction that coincides with the orientation for twinning or cleavage, we believe that facilitated melting along defects, which has been shown in several previous studies (e.g., Tsuchiyama and Takahashi, 1983; Johannes et al., 1994; Incel et al., 2023), is a robust argument for intracrystalline melting.

L406-407: “difficulty to activate plastic deformation” can also be induced by the quite short duration of dynamic slip where (bulk) temperature increase is not high enough to activate plastic deformation. Also, local depletion of water as monitored by local pore pressure drop also plays a role in deactivation of plastic deformation.

A32: We agree and we added: “..., especially within such short durations of dynamic slip of less than a second.” (lines 447-448)

L411: I would suggest rephrasing “amorphous feldspar gouge has no lubricating effect” into “amorphous feldspar gouge has little lubricating effect for short, dynamic slip in intact rocks” because it would weaken if we consider a longer slip or slip on a preexisting fault (like Lockner et al., 2017) as the authors discussed later in the manuscript.

A33: Here, we distinguish between amorphous material due to grinding and amorphous material due to melting. The former appears to have no lubricating effect although amorphous feldspar is abundant as evidenced by the nanostructure of the fault gouge and the mechanical data of self-stabilising sample WG12 (see Figs. 1; 2c, d; 4a, b). To make it clear that we distinguish between different origins of amorphous material, we modified the sentence to: “As feldspar minerals represent the most abundant mineral group in crustal rocks, our observation that nanocrystalline or amorphous feldspar gouge, produced by mechanical grinding (samples WG06 and WG12; Figs. 1; 3; 4), has no detectable lubricating effect, appears relevant to better understand fault dynamics in silicate host rocks of the upper crust” (lines 448-452).

L507: I agree with the authors that melt patches can be formed even during microseismic events. However, I am a bit wondering if a “pseudotachylite” can be formed during a microseismic event as a “pseudotachylite” is often described as a continuous, straight melt in nature. This sentence may be rephrased something like: “even microseismic events ($M_w < 2$) could produce melt patches in granitoid rocks that can be ingredients of pseudotachylites.”

A34: Again, we fully agree and we thank the author for pointing this out. We changed the wording accordingly (lines 579-581).

Figure 3a: Please indicate shear direction. Same for Figures 4, 5, 7. Which side is pushed forward?

A35: The shear directions and the location of the FIB foils are indicated in the overview BSE images displayed in Fig. 2. We find it difficult to indicate the shear sense within the fault gouge at the nanoscale as the FIB sections were cut more or less perpendicular to the direction of shear resulting in the absence of shear indicators within the nanostructures. We hope that the indication of the shear senses at lower magnification exhibited in Fig. 2 will be sufficient.

Figure 3b: Where is this close-up view taken from in panel a? Please indicate.

A36: We added the location of the high magnification image, shown in b, to the overview image Fig. 3a.

Figure 9: What did determine the range of “W” for the dry dynamic case? Please provide an

explanation for this in the figure caption. Also, the unit for dW/dt maybe W/m^2 while it is W/m now in the figure.

A37: Many thanks for pointing out the error in the unit for power, which has been corrected. The range of 'W' in the dry dynamic case results from either using the full shear stress – slip curve to compute W_{tot} (upper bound estimate) or only the fraction of the work that corresponds to pre-peak slip and an idealised slip weakening behaviour from peak stress to a residual strength of 40 MPa with a slip weakening distance of 1.5 mm as in the quasistatic dry test. As requested, we added this information to the figure caption of Fig. 9.

Figure S2f: Please provide a scale bar in the panel.

A38: The location of this image is indicated in Figure S2e, which is provided with a scale bar. We however agree that an individual scale bar might be helpful, and added one as requested.

Reviewer #3 (Remarks to the Author):

Review of “Nanostructures as indicator for deformation dynamics” by Incel et al.

The authors experimentally analyze the relationships between slip dynamics and gouge structure associated with experimental faults. The authors used intact granite samples that were loaded in a triaxial system under conditions of the upper crust with or without fluids. The samples failure by shear along the freshly formed faults occurred under quasi-static, and weak and full dynamic with displacement < 4 mm. The authors correlated the intensity of gouge amorphization with failure style. They found indication of melting and nanograins are common in samples deformed at high rates. They also correlated the nanostructure with power dissipation, energy input, and strength evolution. It is suggested that gouge nanotextures can form even during small slip events of intact rocks.

The authors presented a great piece of research which is professionally written and well organized. The manuscript can be published “as is.” Admittedly, this opinion of mine (Ze'ev Reches) could be biased because the analysis agrees with results of research conducted by my group for years, as outlined below. The comments are divided into major comments (actually they are suggestions) and minor comments.

Ze'ev Reches

Major comments

1. First, my biased view. The current research approach is unique and important. Unlike most studies of earthquake stability, which analyze large displacement between two blocks at frictional contact, the authors started with a solid granite sample that fails and slips under continuous loading. This approach demonstrates the fundamental process of earthquake nucleation and rupture. Admittedly, I made this strong statement as it fits my view of earthquake instability. For example, Reches (1999) concluded: “The nucleation of earthquakes and the associated unstable slip propagation should be analyzed in terms of strength, unstable yielding and rupture of intact rocks.” Muhuri et al (2003) stated: “...that gouge particles are cemented by chemical processes during hold periods and ...the cyclical strength variations are controlled by cohesion strengthening rather than by friction changes.” Recently, Reches & Fineberg (2023) argued “...that earthquake dynamics are best understood in terms of dynamic fracture mechanics and not governed by the frictional properties of faults.” (see: Reches, 1999, EPSL; Muhuri et al., 2003, Geology; Reches & Fineberg, 2023, JGR). This view is based on the ubiquitous observation that the cores of most fault zones are composed of fine-grained crushed rocks, and at typical earthquake depths (elevated temperature and presence of water, e.g., current experiments) such fine-grained material undergoes cementation and fast lithification. Slide-hold-slide experimental

analyses showed that faults can regain their strength during period of hours to weeks (Karner et al., 1997; Muhuri et al., 2003). Similar healing strengthening was documented in-situ after major earthquakes (e.g. After Kobe earthquake in Tadokoro et al., 1999. S wave splitting....JGR 104, 981-991., and Li et al., 1998, Evidence of shallow fault zone strengthening after the 1992 M7.5 Landers, California, earthquake: Science, v. 279, p. 217–219).

A39: Please see our answer A40 below. We now cite Reches (1999) and Reches and Fineberg (2023; line 101).

2. The authors are aware of the possibility of fault zone cementation due to its fine grains (lines 58-61), but the main reason for using intact samples is gouge microstructure of a fresh fault (lines 97-100). While this is a good reason, as demonstrated in the paper, it is suggested to discuss and emphasize the significance of the present approach for characterization of the microstructure associated with failure stability of a cemented, brittle intact fault zone. Fault cementation was studied experimentally with pre-cut samples (above), and the present work goes further and analyzes the effects of a “perfect & complete” lithification on microstructure and failure stability. The authors briefly mention healing and sealing (lines 91-94), but did not discuss the effect of these processes on fault cohesive strength, as it approaches the original strength (Fig. 4 in Reches and Lockner, 2010). I think that the possibilities of a fault to regain its cohesive strength should be discussed in the context of the present analysis.

A40: We fully agree with the reviewer: the use of intact rocks represents an end-member of “perfectly healed/sealed” fault rock, and it is in this spirit that we undertook this study. We however believe that mentioning the importance of cementation processes is best placed in the introduction as it demonstrates the motivation and importance of studying the coupling between fault dynamics and fault gouge structure in initially intact rocks. We added a few sentences to the introduction to further highlight the importance of the restoration of strength through chemical reactions, and now also refer to some publications the reviewer suggested (lines 96-101). In addition, we added statements in the discussion to not only indicate the importance of amorphous material for the onset of melting but also for the cementation processes in fault gouges (lines 553-555).

3. The authors show (lines 27-30; 499-502) that their observations of amorphization and melting do not necessarily require large displacements and/or temperature. This is probably the most important conclusion of this study.

A41: We agree that the observation that “mature” faults do not require large displacements (in feldspar-rich rocks) is indeed remarkable and probably one of the most important conclusions of this study. We therefore modified the text to further highlight this outcome (line 465).

Application of these experimental observations to field observations of small faults is not easy, probably due to reformation of the gouge material. In our analysis of faults in a syenite intrusion (Katz et al., 2003), we analyzed the microstructure of faults with displacements up to 20 mm. The fault zone is composed of fine-grain quartz, opaque oxides, and calcite (Fig. on left, Fig. 11 in Katz et al). While the general shape of this fault is similar to the present experimental faults, the original composition and sub-microstructure of the field fault were altered since its formation (line 59-61), so the slip stability cannot be inferred. See also the description of the gouge of Bosman fault in Wilson et al. (2006).

A42: We thank the reviewer very much for pointing us towards these publications. We are now referring to the publication by Wilson et al. (lines 547).

Minor comments (by line number)

117-125 the procedures of WG06 and WG12 are described too briefly, while understanding the slip rate control procedure of these samples in very central.

A43: We provide further information in lines 601-607, and cite the publication in which the experimental procedure is described in more detail (lines 605-621).

129-131 understood, but pity that this rate was measured...

A44: That is indeed unfortunate.

145-148 this is not “friction coefficients”, replace “shear/normal” ratio

A45: Correct: we used the terminology of “apparent friction”, since the shear/normal ratio is eventually the friction coefficient when the fault is complete (assuming no cohesion), but is not really friction when there is no fault. We made this point more explicit in the text to avoid confusion (lines 150-151). Figure 1 is labelled correctly, as the shear/effective normal stress ratio.

169 , Fig 2g why are these ‘cooling cracks?’

A46: Referring to these structures as cooling cracks produced some confusion. We are therefore no longer making this association.

195-197 important observation

A47: Thank you.

253-256 while this statement is apparently correct, there are only four samples here... so any “trend”

A48: We agree that four tests are probably the minimum number of tests necessary to discuss any trends. The differences in nanostructure among our samples are however so striking that we are confident that there is indeed a trend. Furthermore, our results match very well with observations made in previous studies (e.g., Yund et al., 1990; Lockner et al., 2017; Passelègue et al., 2016, Acosta et al., 2018).

275-282 what are the relationships to previous studies? The 285 Mwm-2 is very high. And it is probably related to the energy dissipated in the extreme conditions at the rupture front (e.g., Reches and Dewers, 2005).

A49: This number is indeed extremely high: it simply results from dissipating the mechanical work over a timescale of 2 ms (instead of 0.5 s for our lower bound estimate), that is a slip rate of the order of 2 m/s (well within what is expected during seismic slip, see Figure 4 of Di Toro et al., Nature 2011). Our point in the paper is however to state that the quantity that changes most between the tests is power, not work, so that it is reasonable to attribute the changes in microstructures to changes in power dissipation. We added a comment in the text to compare with previous data (lines 285-287).

286-288 this is correct, yet note that comminution referred melting as it is easier to melt fine grains. Your line 342.

A50: We agree.

346-349 speculative...

A51: We re-arranged this subsection and these sentences were deleted.

384-395 This is a puzzling part. The piston is the loading system that reacts to stress changed of the sample, thus, unless stress measured ON the sample shows otherwise, one cannot separate the piston from the sample.

A52: In quasistatic systems, stress must equal strength, and the stress in the piston is the same as in sample since they are in series. However, if there is acceleration (as during dynamic slip), the difference between stress and strength is mass times acceleration. The inertia allows the piston to move and unload below the fault strength during deceleration. In other words, the kinetic energy accumulated during acceleration helps the piston move

further, hence unloading more. The details of the mathematical solution of the spring-slider with mass are shown in the supplementary material.

406 the experiments show “fault lubrication”

A53: Please see our answer A54 below.

427-431 do you think that the weakening is due to piston inertia? I do not agree. The dynamic sample failure allows the piston to accelerate and not vice versa.

A54: It is correct to state that the weakening allows the piston to accelerate. Our statement was misleading: we meant to write that “... the large stress drops observed here are found to be explainable by piston inertia and do not necessarily require enhanced dynamic weakening” (lines 469-473).

456-451 also Liao et al., 2014 (Fig.3)

A55: Many thanks for suggesting this publication, which we now cite to strengthen our statement (line 503).

Reviewer #1 (Remarks to the Author):

Dear Editor, Dear authors

This is my second review of the manuscript. It is the revised version following the previous review. The authors did a good job responding accurately to most of the comments. Some issues were not addressed because I did not point them out specifically enough. Fortunately, the other reviewer raised similar comments, which improved the manuscript. The revision has significantly improved the quality of the paper, and it could serve as a guide for future work in this area. I have made several comments and recommend that the manuscript be considered for publication once these comments have been addressed.

We thank reviewer #1 very much for taking the time to go through the revised manuscript again, as well as for the positive feedback, and for the comments.

Comments

Line 1-2. Title: The authors thoroughly document the production of amorphous materials (AMs) and their associated powder density. However, the current title is too general to convey this finding. I suggest highlighting this key finding in the title, as it is highlighted in the manuscript (line 108-115). For example, "Frictional-Original Nanostructures as an Indicator for Deformation Dynamics" or "Gouge Amorphization as an Indicator for Deformation Dynamics." The authors can certainly do better than I can.

Many thanks for these suggestions. We agree that a more precise title is often better. In our case, however, it is not only the amount of amorphous material that acts as an indicator for deformation dynamics but also the partial melting of biotite and orthoclase as well as the growth of magnetite. Due to the presence of several nanostructural indicators, we would like to keep the more general title as it better represents the study, we find, and hope that the reviewer agrees.

Line 259-264. Why is the intensity of the pure AMs different between WG14 and WG N03 (see Fig. 9)? Is it due to the different thicknesses of the FIB samples? Or, is it due to their different chemical compositions (biotite/orthoclase)? Could this affect the statement on powder density calculation?

This is a very good observation. Since the selected area electron diffraction (SAED) was done in areas showing differences in chemical composition, it may be possible that these differences caused the variation in intensity. The thicknesses of the FIB foils were only measured at the top and bottom of the FIB foils. Therefore, we do not have any information about local differences throughout the foils, and can thus not link these intensity variations to any variations in FIB thickness. The important information, however, is that both areas reveal amorphous material as evidenced by the absence of diffraction spots.

Figure 4: The authors have confirmed that the dark mica is biotite, and it should be modified accordingly.

Since biotite belongs to the group of dark mica it is not incorrect to indicate this phase as dark mica. We however followed the suggestions by the reviewer and changed the name from dark mica to biotite in Fig. 4.